# Hydrological Signals in Tilt and Gravity Residuals at Conrad Observatory (Austria)

Bruno Meurers[1], Gábor Papp[2], Hannu Ruotsalainen[3], Judit Benedek[2], Roman Leonhardt[4]

[1] University of Vienna, Department of Meteorology and Geophysics, 1090 Wien, Austria
5  [2] Geodetic and Geophysical Institute, Research Centre for Astronomy and Earth Sciences, Loránd Eötvös Research Network, 9400 Sopron, Hungary
[3] Finnish Geospatial Research Institute, FGI, National Land Survey of Finland, 02430 Masala, Finland
[4] Zentralanstalt für Meteorologie und Geodynamik (ZAMG), 1190 Wien, Austria

*Correspondence to*: Bruno Meurers (bruno.meurers@univie.ac.at)

10  **Abstract.** The Superconducting Gravimeter GWR C025 monitors the time variation of gravity at the Conrad Observatory (Austria) since autumn 2007. Two tiltmeters operate continuously since spring 2016: a 5.5 m long interferometric water level tiltmeter and a Lippmann-type 2D pendulum tilt sensor. The co-located and co-oriented set up enables a wide range of investigations because tilts are sensitive both to geometrical solid Earth deformations and to gravity potential changes. The tide free residuals of the SG and both tiltmeters clearly reflect the gravity/deformation effects associated with short- and 15  long-term environmental processes and reveal a complex water transport process at the observatory site. Water accumulation on the terrain surface causes short-term (a few hours) effects which are clearly imaged by the SG gravity and N-S tilt residuals. Long-term (> a few days/weeks) tilt and gravity variations occur frequently after long-lasting rain, heavy rain or rapid snowmelt. Gravity and tilt residuals are associated with the same hydrological process but have different physical causes. SG gravity residuals reveal the gravitational effect of water mass transport, while modelling results exclude a purely 20  gravitational source of the observed tilts. Tilt residuals show the response on surface loading instead. Tilts can be strongly affected by strain-tilt coupling (cavity effect). N-S tilt signals are much stronger than those of the E-W component most probably due to the cavity effect of the 144 m long tunnel oriented in E-W direction.

## 1 Introduction

The gravity field of the Earth changes temporally mainly because of external forcing, but also due to direct gravitational 25  (Newtonian) and indirect effects of mass transport in the entire Earth system. This happens at all spatial and temporal scales, from local to global and from very short-term to secular. Mass transport does not only change the density distribution, which directly affects the gravity potential, but mostly causes deformation processes due to loading (e.g. Farrell, 1972; Hinderer and Legros 1989). Today, superconducting gravimeters (SG) are the most sensitive instruments for monitoring the temporal variation of the magnitude of the gravity vector. The SG sensor axis is aligned with a plumb line of the gravity field by a tilt 30  compensation system that keeps any misalignment less than 1 µrad (Hinderer et al., 2007). SGs provide highly precise time

series of gravity variations reflecting various geodynamical phenomena like Earth tides, Earth rotation, normal modes, volcanoes and environmental (including hydrological) gravity effects (e.g. Hinderer et al., 2007). Tilt sensors are sensitive to the horizontal component of the gravity vector as well as to rotation of the tiltmeter base and monitor the angle between the sensor axis and the plumb line. Both gravimeters and tiltmeters react on purely gravitational effects caused by

- the Earth's interaction with the Sun and planetary bodies (tides)
- any kind of mass redistribution within the entire Earth system
- Earth rotation changes.

Global geodynamic processes like Earth and ocean tides, normal modes and Earth rotation changes produce global deformation of the Earth while mass movement in the Earth system (atmosphere, hydrosphere, cryosphere, geosphere)
produces global to local deformation due to surface or internal mass loading (atmospheric pressure, hydrological water transport, magma intrusion etc.). The sensitivity of gravimeters and tiltmeters with respect to deformation effects is different. Radial displacement due to deformation results to additional gravity changes, because the sensor moves within the Earth's gravity field. However, as displacement by local load mass is very small, this effect is negligible at local scale (e.g. Llubes et al., 2004) except when inertial acceleration dominates particularly at higher frequencies (Zürn, 2002). Contrary, tiltmeters
are extremely sensitive to even very small deformation. They are able to resolve tilts as small as 1 nrad, which corresponds to a vertical displacement of 1 mm over 1000 km baseline. Figure 1 illustrates how tilts originate depending on material properties of the Earth. Gravitational (Newtonian) tilt is the change of the plumb line direction at the sensor location as it would happen on a non-deformable planet due to the spatial displacement of the equipotential surfaces. The latter is caused either by external forcing fields (tides) or by mass redistribution. Deformation produces tilt if the orientation of the surface
the tilt sensor is mounted on changes with respect to the plumb line. On a non-rigid planet both effects interfere. Deformation is caused by a global stress field (as in case of the body tides) or by loading (atmosphere, water/snow accumulation on surface or below, pore pressure changes etc.). In addition, as well described by Harrison (1976) or Baker (1980), tiltmeter records can be strongly affected by strain-tilt coupling (also called strain-induced tilt) arising from deformation of the cavity in case of underground installations (cavity effect) (Baker and Lennon, 1973; King and Bilham,
1973; Agnew, 1986), surface topography (topographic effect) (e.g. Harrison, 1978), and geological inhomogeneities in the close vicinity (geological effect) (e.g. Kohl and Levine, 1995). These local effects depend on geometry and size of the cavity in which the tiltmeters are installed and on the topography shape. In case of a horizontal tunnel, tilts perpendicular to the tunnel axis will be strongly affected while tilts along the tunnel axis remain widely unaffected (King and Bilham, 1973; Harrison, 1976) provided the tiltmeter is located not too close to the end wall of the tunnel.
SGs show very low instrumental drift of a few nm s$^{-2}$ per year, which can be accurately modeled by linear or exponential time functions (Van Camp and Francis, 2007). In particular since the development of SGs, gravity monitoring has become a valuable tool for hydrogeology investigation applied in very different hydrological settings, complementing the hydrological instrumentation. Gravimeters are very sensitive to mass changes integrated at local scale (e.g. Van Camp et al., 2017). Time-lapse microgravity surveys and SG time series provide useful estimates of water storage changes (e.g. Van Camp et al.,

2006; Davis et al, 2008; Krause et al., 2009; Longuevergne et al., 2009; Creutzfeldt et al., 2010; Lampitelli and Francis, 2010; Hector et al., 2015; Güntner et al., 2017). These techniques have been successfully applied also in karst environment (e.g. Jacob et al., 2009; Fores et al., 2014; Champollion et al., 2018; Mouyen et al., 2019; Watlet et al., 2020).

In contrast, tiltmeter signals dominantly reflect the response on crustal deformation. Tiltmeter observations have widely been used for hydrogeological studies. Herbst (1979) reports tilt signals in the period range of several days obtained from Askania

borehole tiltmeter measurements in Zellerfeld-Mühlenhöhe (Germany) which occurred at precipitation events or during snowmelt periods. He explained the tilt response by lateral fluctuations in the fracture water level inducing pressure differences in adjacent fracture systems, which consequently cause elastic bending of rock structures. Jacob et al. (2010) studied water storage dynamics in the karst area of the Larzac plateau (France). Finite element modelling suggests that deformation due to water pressure changes in fractures is the most reasonable mechanism for explaining observed tilts after

heavy precipitation. Tenze et al. (2012) investigated the effect of underground karstic water flow on tilt observed by two horizontal pendulums in the Grotta Gigante (Italy) and revealed a linear relation between maximum tilt and amount of water entering the Karst system during flood events. Lesparre et al. (2016) interpret tiltmeter observations inside the Fontaine de Vaucluse karst system as infiltration effect of water after rainfall, which changes the pressure in fractures and consequently induces deformation.

Active pumping or injection experiments at different spatial scale have proven high sensitivity of tilt to pore pressure changes (Weise, 1992; Kümpel et al., 1996; Weise et al., 1999; Fujimori et al., 2001; Jahr et al., 2008; Jahr et al., 2018). Within the framework of the large-scale injection experiment at the KTB deep drilling site (Germany), Jahr et al. (2006a, 2006b, 2008) studied the surface deformation due to fluid induced stress changes by borehole tiltmeter array observations. They detected tilt signals with magnitudes between 450 nrad and 700 nrad after three months of water injection and

interpreted the observations as deformation effect extending from the upper crust to the surface, caused by induced pore-pressure changes. Jahr et al. (2009) analyzed high resolution (1 nrad) tilt observations at the Geodynamic Observatory Moxa (Germany) revealing a strong correlation of tilt signals with ground water level changes. All these studies show that pore pressure changes due to water content variations in the subsurface, e.g. as result of precipitation or ground water level variations, can induce tilt.

The Central Institute for Meteorology and Geodynamics (ZAMG, Austria) operates the Superconducting Gravimeter (SG) GWR-C025 since 1995 within the framework of the Global Geodynamics Project (GGP, Crossley et al., 1999) and later the International Geodynamics and Earth Tide Service (IGETS) (Voigt et al., 2016). After terminating a gravity time series at Vienna (Austria) extending over 12 years, the SG was moved to the Conrad observatory (CO, Austria) in autumn 2007, starting a gravity time series over 11 years that lasted until November 2018. Looking at the non-tidal contribution to gravity

variations revealed a much larger hydrological impact on the time series at CO than at Vienna. This is obviously due to complex water infiltration processes taking place after long-lasting rain or rapid snowmelt (Mikolaj and Meurers, 2013), because CO is located in a karst area, where processes are probably even more complicated than in other hydrogeological contexts. Heavy rain and rapid snowmelt cause long-term (a few weeks) residual features the source of which could not be

unambiguously identified so far. The installation of two tiltmeters in 2014 provided new insight into possible scenarios of
hydrological water transport at CO by comparing tide-free SG and tilt time-series, which is subject of the investigation
reported subsequently.

## 2 Observation site and instrumentation

The Conrad observatory is a geophysical/geodynamic research facility located 60 km SW of Vienna (Austria) in a carbonate
region belonging to the Eastern foothill of the Eastern Alps, close to the top of mountain Trafelberg at an elevation of 1050
m. The Trafelberg Mountain itself is part of the Northern Calcareous Alps and shows a complicated nappe-structure
consisting of Main Dolomite and Wetterstein/Gutenstein limestone (Blaumoser, 2011; Bryda and Posch-Trözmüller, 2015).
Three karstic caves are known in the wider surroundings of the observatory (Hartmann and Hartmann, 2000). No natural
springs exist on Trafelberg itself (Deisl et al., 2013). Therefore karstic phenomena like complex underground drainage
systems, karst aquifers, caves and cavern systems, as well as sinkholes are expected to be present. Figure 2 shows the
observatory surroundings. The broad local topography low centred 100-200 m west of the observatory reflects probably a
sinkhole filled by sediments today. Refraction seismic and geoelectric surveys estimate the maximum depth to consolidated
rocks at 30 m (Seren, pers. com.).

The observatory consists of a building (ceiling height about 4 m) for offices/laboratories and a 144 m long and 3 m wide
tunnel drilled in E-W direction (Fig. 3). In one of the laboratories a massive concrete pier is directly connected to solid rock
for gravimeter installations. Prior to building construction, a huge amount of rock was blasted out of the terrain. Before the
concrete foundation plate was made for the building, the remaining cragged and rough rock surface has been levelled by a
gravel sheet. After completion of the building, the space next to and above the building was refilled by the excavated
material in order to restore the original terrain shape. Coverage amounts to approximately 7 m just above the SG. The gravel
sheet below the building is a potential water storage reservoir influencing observed gravity. The tunnel surroundings consist
of solid rocks; the coverage increases towards E from 15 m at the tunnel entrance to about 55 m at the end, with
approximately 33 m at the tiltmeter pier. Given the geometry and orientation of the tunnel, cavity effects are expected
strongest in N-S tilts.

Gravity data are sampled with 1 Hz by two redundant digital volt meters (DVM) for detecting possible long-term scale factor
changes of the DVMs. SG calibrations by co-located absolute gravimeter (JILAg-6, FG-5) observations took place twice a
year, supported by numerous SG/Scintrex CG-5 relative gravimeter intercomparisons (Meurers, 2012; Meurers, 2018a).
Commonly, the SG scale factor (SF) is assumed constant as long as the hardware (e.g. coil geometry, transfer function) does
not change (Goodkind, 1999), which allows for averaging the calibration results (Van Camp et al., 2016; Crossley et al.,
2018). Systematic SF changes, if present and larger than 0.1-0.2‰, are reliably detectable by studying the temporal M2 tidal
parameter modulation of successive tidal analyses over 1 yr intervals. Combining calibration results and M2 parameter

modulation studies (Meurers et al., 2016) proved the accuracy and time-stability of the SG scale factor at CO to be far below 1‰ (Meurers, 2018a).

In August 2014, the Geodetic and Geophysical Institute (GGI Sopron, Hungary) installed a 5.5 m long in one end recording Michelson-Gale type interferometric water level tiltmeter (iWT), designed by the Finnish Geodetic Institute (FGI) (Ruotsalainen et al., 2016a; 2016b; Ruotsalainen, 2018), on a 6 m long pier in the middle of the tunnel, about 94 m far from

the SG. Continuous tilt measurements started at CO in order to monitor geodynamical phenomena like microseisms, free oscillations of the Earth, earth tides, mass loading effects (ocean tidal and atmospheric loading) and possible crustal deformations. In July 2015, a Lippmann HRTM 2D pendulum tilt sensor (LTS) with <1 nrad resolution ([https://www.l-gm.de/en/en_tiltmeter.html](https://www.l-gm.de/en/en_tiltmeter.html)) was installed by GGI close to the iWT on the same pier (Papp et al., 2019). This setup of instruments based on different physical principles (relative height change of a level surface vs. inclination change of the

plumb line) allows for comparing the response of tiltmeters with long (several meters) and short (a few decimetres) base length. While iWT monitors E-W tilts, LTS provides both N-S and E-W tilt time series. The tiltmeter sampling rate is 1 Hz (LTS) and 15 Hz (iWT) respectively. The scale factor of the LTS tiltmeter is factory based. The iWT scale factor is absolute and based on optical interferometry in CO station condition. The iWT tiltmeter detects crustal tilt from water level variations at one end of the tube by interference phase values, which are converted to tilt by a conversion factor based on laser

wavelength, refraction coefficient of water and tube length of the tiltmeter (Ruotsalainen, 2018).

All instruments are underground installations in a thermally stable environment. The tiltmeters are located approximately 33 m below ground surface. Based on theoretical calculations by Harrison and Herbst (1977), Bonaccorso et al. (1999) estimate that the maximum amplitude of thermoelastic tilt of the rocks beneath the surface decays towards zero at 10 m depth. Even if this approach might underestimate the real thermoelastic effect as shown by experiments with shallow borehole tiltmeters at

different depth (Bonaccorso et al., 1999), the coverage of 33 m should reduce thermoelastic tilt deep in the tunnel.

In order to investigate atmospheric and precipitation effects on gravity, a wide range of meteorological parameters are monitored by mobile and permanent sensors:

− air pressure, air temperature, humidity sensors located outside above the laboratory; air pressure sensor included in the SG acquisition system; air pressure, temperature and humidity sensors integrated within the LTS tiltmeter housing; additional

air pressure and temperature sensors in the observatory labs and the tunnel,

− Tipping bucket rain gauge model AP-23 (Anton Paar) with 0.1 mm resolution,

− Disdrometer (Thies) measuring size and fall speed of precipitation particles and classifying the precipitation type by the SYNOP code,

− 3D ultrasonic anemometer (Thies).

− SSG-2 snow scale (Sommer Company, Austria) monitoring the weight of the snow pack in front of the observatory and providing snow water equivalent data. The snow scale was out of operation between January 1, 2018 and March 15, 2018. Missing data has been replaced by information from a nearby (150 m SW of the observatory) snow height sensor.

## 3 Gravity and tilt pre-processing and determination of residuals

For separating small amplitude gravity and tilt signals of different physical origin like e.g. hydrological response or tectonic signals we need to subtract tidal effects, which dominate the gravity and tilt time series. The atmospheric pressure and polar motion are also known to contribute remarkably to temporal gravity and tilt variations although much less than the tides. Both the SG and the tiltmeters are relative instruments and hence may exhibit instrumental drift. Generally, the SG drift is expected to be only a few $\mathrm{nm\,s^{-2}}$ per year. Absolute gravity observations performed at CO did not reveal significant instrumental drift of the SG until now (Meurers, 2018b). However, the tilt sensors show strong drift dominated by linear trends up to $-10$ µrad yr$^{-1}$ and $+2.5$ µrad yr$^{-1}$ for the LTS and iWT sensors respectively and by possible thermal origin. Therefore, the gravity and tilt time series must be properly processed for deriving residual time series. Pre-processing and determination of gravity/tilt residuals followed the procedure which is standard for SG time series (Hinderer et al., 2007). For decimating 1 Hz samples to 1 min or 1 h samples we applied numerical filters g1s1m and g1m1h respectively (http://www.eas.slu.edu/GGP/ggpfilters.html). Local tide models in the diurnal and sub-diurnal frequency bands as well as air pressure admittances were derived individually for each sensor from tidal analyses by applying ETERNA v3.4 and ETERNA-x et34-x-v80 (Wenzel, 1996; Schüller, 2020). Tidal parameters of theoretical body tide models (e.g. Dehant et al., 1999) are used for long-period tides. The following pre-processing steps had to be applied additionally for the tilt sensors:

− Interpolation of 15 Hz iWT data to 5 Hz samples and decimation of 5 Hz data to 1 Hz samples by using a Gaussian operator with 61 coefficients equivalent to 1 min time length.

− Correction of transient signals due to thermal disturbances in the tunnel, which are very small but happen occasionally during maintenance work. Until August 2017, episodic temperature increase of a few 0.01°C inside the LTS was observed by the built-in sensor, generating tilt signals much larger than the tidal signal. The temperature correction was based on linear or nonlinear models depending on the thermal event. Since August 2017 both tilt sensors are isolated from the temperature fluctuation in the tunnel by styrox-plate insulation around the tiltmeters, which effectively suppresses the thermal disturbances.

− Correction of steps, in particular for iWT data, by applying TSOFT (Van Camp and Vauterin, 2005) and own codes. Due to its incremental measuring principle iWT sometimes suffers from interference phase cycle-slips; the correct interpretation of the interferogram phase fails if the phase change between two consecutive interferograms is larger than one interference phase value of typically 203.6 nm. This happens during large earthquakes when ground motion is too fast so that the fluid level of the instrument cannot follow fast and large seismic surface wave arrivals in the first minutes.

− Removing low order polynomial trends.

## 4 Local tide models and air pressure admittance

The local tide model for gravity matches the theoretical body tide models (e.g. Dehant et al., 1999; Mathews, 2001) and the ocean tide loading predictions provided by Bos and Scherneck (2017) almost perfectly (e.g. CSR4.0 (Eanes 1994), GOT00.2 (Ray 1999), TPXO7.2, TPXO9 (Egbert and Erofeeva 2002), FES2004 (Lyard et al. 2006), EOT11a (Savcenko and Bosch 2011), DTU10 (Cheng and Andersen 2010), HAMTIDE (Taguchi et al. 2014) and NAO99 (Matsumoto et al. 2000)). This is due to the high accuracy of both the SG scale factor determination (0.2 ‰) at CO (Meurers, 2018a) and the tidal analysis which is based on time series longer than 10 yr (Meurers, 2018b). The formal errors of gravimetric factors are far below 0.1 ‰ for the main tidal constituents. The RMS error of a single observation estimated from the tidal adjustment residuals, which were calculated by using the adjusted tidal parameters, is 0.6 nm s$^{-2}$ or 0.9 ‰ of the tidal peak-to-peak M2 amplitude only.

Local tide models for the tilt sensors are much less accurate. The RMS errors of a single observation derived from the LSQ adjustment of tidal parameters range from 1.6 to 2.9 nrad, which is in the order of about 2-4 % of the peak-to-peak M2 tidal signal. Also, much less data (LTS N-S: 21700 hourly samples within 1064 days; SG: 83500 hourly data within 3512 days) could be used for tidal analyses. Table 1 compares the tidal parameters of the main tidal groups for the LTS and iWT tilt sensors. The LTS N-S component turns out to be heavily disturbed by non-tidal excitation, particularly in the diurnal band, while the E-W components do not considerably deviate from the body tide predictions. We also analyzed the data a-priori corrected for atmospheric and induced non-tidal oceanic loading contributions (Boy et al., 2009) provided by EOST loading Service (http://loading.u-strasbg.fr/). After correction, the non-tidal tilt anomaly in the diurnal band still persists. However at CO, ocean loading corrections based on the TPXO9 model (http://holt.oso.chalmers.se/loading/) do not essentially reduce the deviation of observed tilt-factors from the body tide predictions. Because the tunnel axis is oriented in E-W direction, the N-S component corresponds to the tilt perpendicular to the tunnel axis and therefore is extremely sensitive to cavity effects (King and Bilham, 1973; Harrison, 1976; Agnew, 1986). This is the most likely reason for anomalous tidal parameters in N-S tilt, particularly in the diurnal band where tidal N-S tilt wave amplitudes are small (<5 nrad). The high LTS/iWT ratio of the E-W tilt factors hints to calibration errors. LTS tilt factors are by 6-11 % higher than those of the iWT, i.e. the tidal parameters are probably affected also by unknown transfer functions of the tilt sensors. However, we cannot exclude that cavity effects play a role as well, as the respective tilt sensors are not exactly at the same place and have different base length. In order to consider all these problems properly, sensor dependent tidal models have been used for the tilt residual determination.

Air pressure has also a strong impact on observed tilts dominantly due to surface loading (e.g. Rabbel and Zschau, 1995) and directly results to surface and subsurface deformation depending on the spatial scale of load masses (e.g. Llubes et al., 2004). Air pressure changes are caused by air packages with different density and spatial extent passing the station. Therefore, air pressure signatures in tilt time series are expected to be frequency dependent as it is well known from gravity records. Loading by accumulated water or snow produces deformation in a similar way. Hence it is worth to study the air pressure

admittance function for the tilt. Air pressure tilt admittances for tidal frequencies were calculated in a joint adjustment together with the tidal parameters by ETERNA-x et34-x-v80 software (Schüller, 2020). The results in Table 2 represent the diurnal and semidiurnal frequency band only because long-period tides were not included in the adjustment. To get higher frequency information we investigated the frequency dependence of the air pressure admittance by applying cross spectral analysis (Bendat and Piersol, 2010) on several detided tilt time series covering intervals between 2 and 21 days (10 days on average), both for LTS N-S and LTS E-W. For LTS N-S, the air pressure admittances confirm the number resulting from the tidal analysis (Table 2) obtained in the diurnal and semidiurnal frequency band. Clear time variability is seen at frequencies beyond 0.3 mHz (equivalent to a period of about 1 h), which is of instrumental origin. Therefore, separating physically meaningful signals from instrumental artefacts is not possible in the frequency range larger than 0.3 mHz. Details are provided in appendix A. However, at long periods the air pressure signal in the tiltmeter time series is due to geophysical/geodynamical reasons which are probably dominated by deformation due to air pressure loading. Here, the admittance is again much higher for N-S tilt than for E-W tilt (similarly as shown in Table 2) as expected due to the cavity effect. We come back to this in chapter 5.1 when we discuss tilt response to water mass load on the terrain surface.

## 5 Gravity and tilt residuals at Conrad Observatory

Figure 4 presents the final gravity and tilt residuals of the common observation period extending from end of April 2016 until mid of November 2018. Comparing the residuals with cumulative rain and snow (water equivalent) shows an obvious link between both short- and long-term residual anomalies related to different hydrological processes.

### 5.1 Short-term signatures (water accumulation phase)

Figure 5 presents a typical example of a heavy rain event on July 11, 2016. The SG residuals decrease sharply and exactly at the time when rain starts. This is mainly due to the Newtonian effect of rainwater distributed at the terrain surface, above the instrument. Actually, due to their high precision SGs reveal these effects not only in case of heavy rain events but also in case of light rainfall even smaller than 1 mm h$^{-1}$. The gravity residual drop can be very well estimated by multiplying the cumulative rain with a rain admittance factor based on a digital terrain model in high spatial resolution (Meurers et al., 2007). The rain admittance depends on terrain geometry, SG sensor location and on the area of rainwater accumulation. At CO, the rain admittance varies between $-0.26$ nm s$^{-2}$ and $-0.29$ nm s$^{-2}$ per 1 mm rain for accumulation areas between $10^4$ and $10^2$ km$^2$ (Fig. 6, left panel). Correcting for the Newtonian effect of cumulative rain removes the gravity response to rain almost perfectly (Fig. 5, light red line). Of course, the rain admittance concept works only during the accumulation phase while it fails when the residuals recover their initial level after rainfall.

The same approach can be applied for estimating the Newtonian tilt effect of rain water both in N-S and E-W direction. Corresponding rain admittances turn out to be as small as $-1.3 \cdot 10^{-3}$ nrad per 1 mm rain for N-S tilt and $-7.6 \cdot 10^{-3}$ nrad per 1 mm rain for E-W tilt respectively, if the rainfall area extends to more than 2 km symmetrically around the tilt sensor (Fig. 6,

right panel). In case of a rain front the Newtonian effect can be considerably larger and depends on the direction from which the rain front approaches the station. The effect of asymmetric rainfall areas extending to a line just passing the tilt sensor location provides the maximum estimate, which does not exceed $\pm 7.7 \cdot 10^{-2}$ nrad per 1 mm rain at CO. However, in realistic weather situations, the rain to tilt admittance is much smaller and depends on the velocity at which the rain front moves over the sensor. Given these small numbers, the Newtonian tilt effect of rain water or snow turns out to be negligible at CO, because it is far below the reliable resolution of tiltmeters.

Nevertheless, there is a clear and instantaneous N-S tilt response on rain (exemplarily shown by Fig. 5), which is visible in almost all (71 out of 74) heavy rain events. Tilt response on air pressure changes can be ruled out as a reason because the temporal patterns of air pressure and tilt are totally different in most cases, while tilt and cumulative rain match each other. Similar as in case of gravity, we do not observe any time delay between cumulative rain and tilt response. In contrast, tilts in E-W direction rarely show short-term signatures that could be related to rain. In only 10 out of 48 rain events a slight transient residual decrease is visible, which, however, often starts much earlier than rain. Figure 7 (left panel) shows the observed total N-S tilt offsets as function of cumulative rain or of the surface pressure load exerted by cumulative rain at the end of the respective rain event. The average rain admittance results to 0.73 nrad mm$^{-1}$, which is about 580 times larger than the value estimated for purely gravitational tilt (Fig. 6) or about 7.4 nrad hPa$^{-1}$ after converting cumulative rain into surface load pressure. This corresponds to the air pressure admittance for the N-S tilt at about 0.3 mHz. Also, we find a close relation between the response of N-S tilt and gravity to short-term water accumulation at topography (Fig. 7, right panel). The air pressure admittances for the E-W sensors are much weaker than those for the N-S tilt sensor at all frequencies, which may explain why we rarely see E-W tilt effects due to rain: surface load (either due to air pressure or rain/snow) does rarely produce clear signatures in the E-W tilts because the cavity effect is much smaller for E-W tilt than for N-S tilt. Tilt response to surface load by water accumulation evidently compares well with the tilt response to atmospheric pressure changes, both for the N-S and the E-W components. The SG reflects mainly the gravitational effect of the rain/snow water while the deformation effect on gravity (vertical displacement) at the given spatial scale is too small to be detected; contrarily, the tiltmeter responds to deformation caused by the pressure the water exerts onto the terrain surface, similarly as in case of air pressure variations. It is probably the cavity effect, which amplifies the observed tilt such that it emerges from the noise in case of the N-S component, which is oriented perpendicular to the tunnel axis at CO.

The findings above hold also in case of solid precipitation as shown in Fig. 8, which presents an example of gravity and N-S tilt response to pure snow accumulation. Disdrometer data (Fig. 8, coloured dots) show that almost no liquid precipitation is involved. The disdrometer provides information on the aggregate state of the precipitation particles even for extremely little precipitation. However, as indicated by the rain data (Fig. 8, magenta colour), liquid rain does not essentially contribute to water accumulation in the presented case study. Consequently, no essential water infiltration can take place because most precipitation is solid and air temperature remains slightly below the melting point (Fig. 8, green line). Note that heated rain gauges often report solid precipitation incorrectly and/or delayed in time because the solid particles have to melt before they are counted by a bucket rain gauge. The disdrometer detects precipitation starting as snow grains and light drizzle during

night and early morning with an intensity which is too small to be observed by the rain gauge. Precipitation continues as light to heavy snow from 8 UTC onwards. The snow scale indicates the onset of snow cover increase at about 12 UTC. Gravity residuals start decreasing at the same time and reach a local minimum at about 22 UTC when heavy snow fall terminates. The prediction of the cumulative precipitation effect by applying the rain admittance removes the gravity residual drop almost perfectly (Fig. 8, light red line). A significant signal associated with the main snow accumulation phase is also visible in the N-S tilt residuals, comparable in magnitude to rainfall events (compare to Fig. 5), i.e. snow does affect tilts similarly as in case of rain. Snow water equivalent matches the tilt time pattern if properly scaled (Fig. 8, orange line). Again, gravity and tilt react instantaneously i.e. without time delay.

The short-term residual anomalies can therefore be well explained by the accumulation of precipitation on the terrain surface and in the adjacent topsoil. While the gravity response reflects the gravitational acceleration of accumulated water/snow mass, the N-S tilt response is interpretable as pure deformation effect caused by the pressure the water mass exerts on the terrain surface. Similarly as in case of atmospheric pressure changes, the cavity effect enhances observed tilts in N-S direction much more than those oriented E-W. If the accumulation phase is short as in the case studies discussed so far, we do not expect considerable water percolation into the subsurface changing the pore pressure there.

## 5.2 Long-term signatures (water percolation phase)

It is common to most rain events that after rainfall a slow discharge process brings the gravity residuals back to their initial level (Fig. 4). However, in some events the residuals exceed the initial level remarkably, in particular after long-lasting rain or rapid snowmelt. We interpret this as response to downwards water flow (infiltration) from terrain surface into the ground until water is stored somewhere below the SG sensor. This process probably starts as soon as the subsurface is sufficiently saturated by rain or snowmelt water and therefore needs a certain threshold to be triggered. Mangou (2019) estimated that about 20 mm water accumulation within the past 3 days is required. However, this number is a rough estimate. The degree of saturation as well as meteorological conditions (e.g. evaporation rate etc.) plays a role as well.

Interestingly, all these events are associated with simultaneous long-term tilt anomalies: Almost at the same time when the SG gravity residual starts to increase, we see strong signals in the tilt time series as well. N-S tilt shows always a steep residual drop, the E-W tilt residuals (in particular LTS) increase temporarily but with much less amplitude. E-W tilt signals are often masked by noise. All events where we identified long-term signatures both in gravity and tilt residuals are marked by dotted vertical lines in Fig. 4. Figure 9 exemplarily zooms into a long-lasting rain event (Fig. 9, left panel) and into a rapid snowmelt event (Fig. 9, right panel). Once N-S and E-W tilts have reached their extremes they return to their former level; a process which takes about 14 days or more. The short-term signals discussed in 5.1 are visible in Fig. 4 too, even though very small compared to the long-term signal. The long-term anomalies start when sufficient water has percolated downwards into the subsurface either after heavy/long-lasting rainfall or in case of rapid snowmelt. Quantifying the long-term anomalies is not easy because the tilt/gravity response to long-term water transport depends on the overall subsurface saturation for which we have no constraints based on observations. However, there is a significant relation between the long-

term residual anomalies observed in the tilt and gravity residuals (Fig. 10, left panel). Tilt residual anomalies have always either negative (N-S tilt) or positive (E-W tilt) sign. The absolute value of the anomaly amplitudes increases with the amplitude of the gravity residual anomaly, whereby the N-S residual anomaly amplitude is about 7 times larger on average than that of E-W residuals (Fig. 10, right panel).

## 6. Discussion

There are a few candidates for water storage volumes at CO:

− the gravel layer below the concrete foundation plate of the underground observatory building and the laboratories in front of the tunnel,

− fissures and cracks in the solid rock or

− perhaps a karstic volume filled by water after heavy rain/snowmelt.

We first investigate whether a locally limited surface or subsurface mass is able to produce the observed long-term tilt/gravity residuals. Comparing the E-W and N-S tilt data, the amplitude ratio of the long-term residual anomalies turns out to be about −0.15 on average (Fig. 10, right panel). E-W tilt is always positive, N-S tilt is always negative (Fig. 10, left panel). If the observed tilt is solely due to gravitational attraction by a volume of stored water, then the source must be located on a line with azimuth of about 170°. Based on the high resolution digital terrain model (DTM) of the area (Meurers et al., 2007) the existence of any surface depression capable to cumulate run-off water mass (Kalmár and Benedek, 2018) enough for generating the observed tilts can be checked. Figure 11 shows that there are two local topographical lows (valleys) along the profile. However, due to their distances to the observatory enormous amount of water ($> 10^5$ m$^3$) should be accumulated in a corresponding cell of the DTM (determined by the azimuth) to generate even a fraction (1 nrad) of the observed tilts (up to ~1000 nrad). Regarding the horizontal extension of such a cell (50 m x 50 m) 40 m water height would be required to provide this volume. The same holds for a fictitious topographical reservoir located in the very close vicinity ($< 50$ m) since about 1000 m$^3$ water is necessary for the same tiny (1 nrad) gravitational tilt. This volume of water is supplied by 1 mm rain fallen on 1 km$^2$ but obviously even this amount cannot be caught and concentrated near to the observatory as one can conclude from Fig. 2 showing the elevation contour lines. The estimations above are based on forward gravitational modelling of the horizontal attraction of mass columns (e.g. Papp and Benedek, 2000) representing the water mass placed on the top of the topographic mass columns. However, there is no any evidence of such a large basin next to CO in the required azimuth. Another point source modelling shows that along this azimuth no spherical volumes representing one single subsurface cavity partially or completely filled by water would simultaneously explain both gravity and tilt residuals of the events shown in Fig. 4.

Therefore, regarding the long-term residual variations, a pure Newtonian effect of one single source (e.g. one single karstic cave filled by water) representing the water accumulation near the gravity and tilt sensors can be ruled out because of two reasons:

− model calculations show that contrary to the short-term anomalies no reasonable solution exists to explain the observed long-term tilt and gravity effects and

− the onsets of the long-term residual features in gravity and tilt do not coincide exactly in time.

Deformation by increasing pore pressure after water infiltration into the subsurface is the most reasonable explanation for observed tilts. Actually, the observed long-term N-S tilt response (Fig. 4, Fig. 9) is very similar in shape to the observations

reported by Herbst (1979) or by Jahr et al. (2006a, 2006b) in one of the tilt records of a borehole tiltmeter array established at the KTB deep drilling site (Germany).

Certainly, the hydrological water transport process is very complex at CO. Due to the high sensitivity and extremely low and almost linear instrumental drift of SG sensors the SG gravity residual reveals very clearly the Newtonian effect (vertical component) of water mass transport involved in hydrological charge and discharge processes. We modelled the gravity

effect by a simple layer in order to estimate the maximum observable gravity residual drop as function of the layer thickness and of the degree of initial soil/rock saturation. The upper layer boundary coincides with the terrain surface; the lower boundary is defined by shifting the terrain surface vertically downwards. The topography is represented by the same DTM with high spatial resolution, in particular in the vicinity of the SG, as already has been used for the rain admittance calculations. The effective layer density $\delta\rho$ results from Eq. (1):

$$\delta\rho = \phi(S - S_0)\rho_w \tag{1}$$

with $\rho_w$ and $\phi$ denoting water density and rock porosity respectively. $S_0$ and $S$ describe the saturation of the pore volume before (initial saturation) and after downward water mass transport. The model takes into account that water storage is impossible within the volume occupied by the observatory building/tunnel, the foundation plate of the building and the gravimeter pier. Figure 12 shows the modelled gravity as function of layer thickness for $S = 1$ and different degrees of initial

saturation $S_0$, assuming a porosity of $\phi = 0.1$. The effective layer density is 100 kg m$^{-3}$ for initially completely dry rock ($S_0 = 0$). Alternatively we can interpret Fig. 12 also as gravity effect of the same layers as function of layer thickness but for different porosity, assuming $S = 1$ and $S_0 = 0$. Then, the layer density provided in the legend of Fig. 12 translates into porosity after division by 1000. The minimum gravity residual occurs at a layer thickness of about 9 m in each case whereby the drop amplitude increases with decreasing degree of initial saturation $S_0$. Given the terrain model geometry at CO, the

minimum residual drop amplitude never exceeds about 200 nm s$^{-2}$ ($S = 1$ and $S_0 = 0$) at the SG site. However, observed numbers are much smaller. For all events shown in Fig. 4 the residuals never drop by more than about 10 nm s$^{-2}$, which indicates high degree of initial subsurface saturation or porosity lower than assumed in the model. We investigated the period from July 11 and July 14, 2016, (Fig. 9, left panel), during which a series of consecutive heavy or long-lasting rainfall events occurred, in more detail. Simultaneously with the first rainfall on July 11, 2016, the gravity residuals decrease by

about 8 nm s$^{-2}$ and remain nearly at this level after rain has stopped. More heavy rain events follow separated by a couple of hours (Fig. 13, left panel). The residuals always drop instantaneously at the onset of each event but start to increase short time later. In total they increase to a much higher level than they started from at the beginning, although more and more rain

is accumulated. We developed a time-lapse model and compared the time-dependant model response with observed gravity residuals. Unfortunately we cannot constrain our model by hydrological observations. Therefore the very simplistic model is based on following assumptions:

- Constant porosity of $\phi = 0.1$ and constant degree of saturation $S = 1$, which translates into a subsurface density $\delta\rho$ according to Eq. (1). That means water percolating downwards fills the pore volume completely. The choice of the porosity seems to be reasonable. Jacob et al. (2009) report values between 0.04 and 0.12 in a karstic environment (Larzac Plateau, France).

- Rain water is assumed to percolate into the subsurface as a layer of spatially constant thickness $H(t)$. The upper layer boundary coincides with the terrain surface as before while the lower boundary results from shifting the terrain surface vertically downwards.

- The subsurface is partially saturated with degree of saturation $S_0$ at the beginning, i.e. before the rain series starts.

- Based on the mass conservation principle the model keeps the balance between accumulated water $h_w(t)$ and the water percolated into the subsurface. This defines the thickness of the water layer $H$ as function of time $t$:

$$H(t) = \frac{\rho_w}{\delta\rho} h_w(t) = \frac{h_w(t)}{\phi(S-S_0)} \quad \text{or} \quad H(t) = \frac{h_w(t)}{\phi(1-S_0)} \quad \text{for } S = 1 \tag{2}$$

where $h_w(t)$ denotes cumulative rain and $t = 0$ the beginning of the first rain event.

- Water cannot be stored below a maximum layer thickness $H_s$, but disappears from there due to any run-off process. This constrains the maximum level the gravity residuals can ever reach.

- If the layer thickness is less than $H_s$ by the end of the rain event series, the lower boundary continues propagating into depth until the layer thickness has reached $H_s$. However, now the layer thickness increases at the expense of layer density (or of saturation $S$) in order to conserve the total water mass. This assumption considers the general characteristics of the relation between long-lasting rainfall or heavy rain and residual gravity: gravity residuals drop down at first as expected for underground installations, but later start to increase and continue increasing even after rain has stopped at the end of a rain event or rain event series. Figure 9 (left panel) provides a typical example.

Figure 13 (left panel) shows the modelled gravity response to a real cumulative rain function, monitored between July 11 and July 14, 2016, for different degree of initial saturation $S_0$. All models with $S_0 \leq 0.9$ clearly fail, as they are not able to explain the overall gravity residual increase during the rainfall series. Best results are provided for $0.94 \leq S_0 \leq 0.96$ with a model misfit (standard deviation) ranging between 3 nm s$^{-1}$ ($S_0 = 0.95$) and 5 nm s$^{-1}$ (Fig. 13, right panel). For models assuming $S_0 \leq 0.9$ the misfit standard deviation increases to about 19 nm s$^{-1}$. If the subsurface is initially dry ($S_0 = 0$), then the model response (Fig. 13, dark green line) is almost identical with the gravity effect of cumulative rain calculated by applying the admittance concept (Fig. 13, orange line), i.e. all water remains concentrated close to the surface for long time. The key point is that the lower layer boundary has to propagate downwards fast enough to store water below the SG sensor. The model is sensitive to the choice of input parameters like porosity $\phi$ or layer thickness $H_s$. We get the same $H(t)$ as long as the denominator is kept constant in Eq. 2; i.e. we can play off $S_0$ against $\phi$. For example, the choice of $\phi = 0.05$, which is

still reasonable for limestone environment, and $S_0 = 0.9$ would not change the model response. However, if $S_0$ is 0.5, then porosity has to be 0.01, which is very low. Of course, we have to emphasize the simplicity of the model, which, for example, does not allow for horizontal water flow (e.g. Krause et al., 2009) or a direct transport downwards along specific flow paths as expected in karst. Nevertheless, these model results indicate that the saturation seems to be high (>0.9) or porosity to be low at CO. Note that the model implicitly contains the sinkhole SW from the observatory, at least partly. Based on the results from refraction seismic and geoelectric measurements, 3D modelling predicts additional gravity increase of only about 4 nm s$^{-2}$, if porosity of 0.3 and full saturation is assumed for the sinkhole filling. However, this small effect does not change the conclusion drawn from Fig. 13.

Contrary; the tiltmeters are not able to capture the gravitational tilt effect because the latter is too small and thus hidden in the noise. However, in particular the N-S tilt residuals show significant both short- and long-term anomalies, which are associated with the same rain or snowmelt events and are clearly related to the residual patterns captured by the SG gravity record. Therefore we explain the tilt residual anomalies by surface or subsurface deformation. Here we can distinguish two hydrological processes:

− Charge process: Deformation caused by surface load (rain water, snow) produces short-term tilt anomalies associated with heavy precipitation.

− Discharge process: Deformation probably caused by pressure changes in the adjacent fracture system induces long-term tilt anomalies lasting over up to 3 weeks.

In both cases, tilts in N-S direction are enhanced due to the cavity effect. These hydrological processes, either water accumulation at terrain surface (short-term) or subsurface infiltration (long-term), link gravity and tilt residual anomalies. Gravity and tilt respond to these processes based on different physical phenomena: gravitational effects of moving water mass (gravity) vs. deformation due to loading (tilt). The cavity effect enhances the tilt component perpendicular to the tunnel axis due to strain-tilt coupling. Presently, it is not yet clear, if karstic phenomena play an important role at CO as well. No large caves are known in the rock massif the CO is located on. However, we cannot exclude that deformation by internal loading could take place, e.g. when an eventually existing cave or drainage system would be filled by water during hydrological discharge (e.g. Tenze et al., 2012).

## 7. Conclusion

Gravimeters provide the integral effect of water storage changes. The distinct gravity residual anomalies after heavy or long-lasting rain and snowmelt have been observed at CO for long time, and their reason was unclear. Very local water storage just below the observatory building after rapid flow of surface water through the backfill material on top and besides the observatory was the preferred explanation so far. The tiltmeter instrumentation initially established for completely different research goals has brought new insight to the water transport processes at CO. The close link between the long-term gravity and tilt residual anomalies indicates that the discharge process takes place in much larger spatial context. Simplistic models

of uniform water infiltration are able to explain the observed gravity residual increase following heavy or long-lasting rain. Stepping into even more complex quantitative modelling certainly requires full hydrological equipment (soil moisture, ground water etc.) in order to constrain the models. Complementary geophysical investigations like 4D geoelectric monitoring (e.g. Watlet et al., 2018) and cross-correlation of ambient seismic noise, both of which can provide further information on temporal water saturation changes (Fores et al., 2018), are promising techniques for future investigations.

460

**Appendix A: Analysis of the air pressure admittance function for tilt**

Clear time variability is seen in the air pressure admittance function for tilt at higher frequencies, which is obviously related to maintenance work (Fig. A1). In May and September 2018 factory repairs by the manufacturer were necessary after thunderstorm strikes, partly damaging some electronic parts inside the LTS sensor box. Before the LTS repair in May 2018, both admittance and phase increase slightly towards higher frequencies up to 0.3 mHz in all time series (Fig. A1, blue and green lines). Beyond about 0.3 mHz the admittance increase gets much stronger before it drops down at about 3–4 mHz. Note that the admittance functions are not corrected for the unknown transfer functions of the involved sensors. After the first repair, the admittance gets flat or even decreases already at frequencies >0.3 mHz (Fig. A1, yellow and red lines). With very few exceptions, coherence is between 0.6 and 0.8 at frequencies <0.1 mHz for all LTS N-S time series and drops down to 0.3 to 0.4 at higher frequencies. Coherence measures the accuracy of the input/output model and can be derived from the autospectral and cross-spectral density functions (Bendat and Piersol, 2010) of tilt and air pressure. The coherence decreases to less than 0.1 already between 0.1 mHz and 1 mHz after repair in May 2018. The picture is much less clear for LTS E-W. Coherence is at a very low level of 0.1–0.2 at all frequencies indicating that generally no or little dependence on air pressure exists as also suggested by Table 2. Nevertheless, we see a similar admittance change related to sensor maintenance as for LTS N-S.

Tiltmeters are very sensitive to temperature changes. Klügel (2003) revealed instrumental effects of LTS tiltmeters and interpreted them as being caused by quasi-adiabatic temperature changes associated with rapid air pressure variations. For the LTS tiltmeters at CO, the temperature coefficient estimated from disturbances during maintenance work in the tunnel ranges from 3.9 to 4.9 µrad $K^{-1}$ (LTS N-S) and 3.0 to 4.0 µrad $K^{-1}$ (LTS E-W). Typically, rapid air pressure changes caused by convective meteorological events amount up to 3 hPa. Klügel (2003) estimates the temperature variation due to air pressure change at 1.5 mK $hPa^{-1}$. Assuming this number to be valid also for the LTS tiltmeter at CO, air pressure change of 1 hPa translates into a temperature variation up to 4.5 mK and consequently into about 7 nrad tilt. This corresponds to the air pressure admittance at about 0.3 mHz for N-S tilt (Fig. A1). The temperature change itself is below the recording resolution of the LTS temperature sensor (0.01 K). Therefore we did not directly observe temperature signals related to rapid air

pressure changes. A temperature sensor operating close to the end of the tunnel with 2 mK resolution since mid of 2018 indicates indeed a relation between air pressure and temperature change. However, the currently available data do not allow quantitative analyses. Air pressure patterns rapidly passing the station will be seen as high frequency signatures in the air pressure time series; the faster the passing velocity the higher the frequency will appear. This might be the reason for the admittance increase towards higher frequencies observed before the repairs.

Figure A2 proves that admittance function changes over time are of instrumental origin concerning either the tilt or the air pressure sensor or even both. We show the temporal variation of admittances and phases at 4 selected frequencies calculated with air pressure data acquired by the in-built sensor (LTS, Fig. A2, bottom panels) and the air pressure sensor of the SG (Fig. A2, middle panels) respectively. The transfer function of the air pressure sensor is unknown and presently cannot be determined without interrupting the tilt time series. Atmospheric admittance investigations of the SG performed so far revealed the SG air pressure sensor to be stable. Therefore the SG air pressure sensor may serve as reference and we present the temporal admittance function changes of the two pressure sensors in the top panels of Fig. A2. A rapid but steady change happens between March and August 2017. The installation of the thermal insulation in August 2017 did obviously not affect the admittance function. However, after repair in May 2018, a sudden change of admittance and phase is visible at frequencies larger than 0.1 mHz, which is probably related to the maintenance work. The described events appear synchronously in both air pressure to tilt admittance functions independent of the pressure sensor used for evaluation. This suggests that tilt signals with frequencies >0.3 mHz are of instrumental origin or at least strongly affected by instrumental issues after maintenance/transport.

**Code and Data availability**. ETERNA-x et34-x-v80: http://ggp.bkg.bund.de/eterna/; ETERNAv3.4: International Center for Earth Tides, https://webdevel.upf.pf/ICET/home.html; TSOFT: Royal Observatory of Belgium, http://seismologie.oma.be/en/downloads/tsoft; gravity data: International Geodynamics and Earth Tide Service (IGETS), http://igets.u-strasbg.fr/; tilt data are available on request.

**Author contributions**. Tiltmeter set-up and maintenance: GP, JB, HR, RL; SG maintenance: RL with co-workers, BM; processing of tilt data: GP, JB; processing of the gravity data: BM; tidal analyses and cross spectral analysis: BM; final residuals: BM, GP; data analyses: BM, GP, JB; interpretation and discussion of results: all authors; paper writing: BM with contributions from all authors.

**Competing interests**. The authors declare that they have no conflict of interest.

**Acknowledgements.** Both the iWT and the LTS instruments were purchased from the budget dedicated to the development of infrastructure of the Geodetic and Geophysical Institute Sopron, Hungary based on the kind decision of Viktor Wesztergom, director of GGI**.**

The research described in the paper was basically supported by NKFIH-OTKA under the contract K-128527. The great help of ZAMG and its observatory team providing excellent research facilities is gratefully thanked by the authors. The excellent technical support of Frigyes Bánfi, Tibor Molnár and Csaba Molnár (GGI Sopron) is acknowledged too.

We are very grateful to Michel Van Camp and an anonymous reviewer for essential suggestions improving the paper.

At last but not least special thank to Ing. Erich Lippmann for his magnanimous help in the proper maintenance and servicing of his tiltmeter and for his all time readiness for consulting and discussion.

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

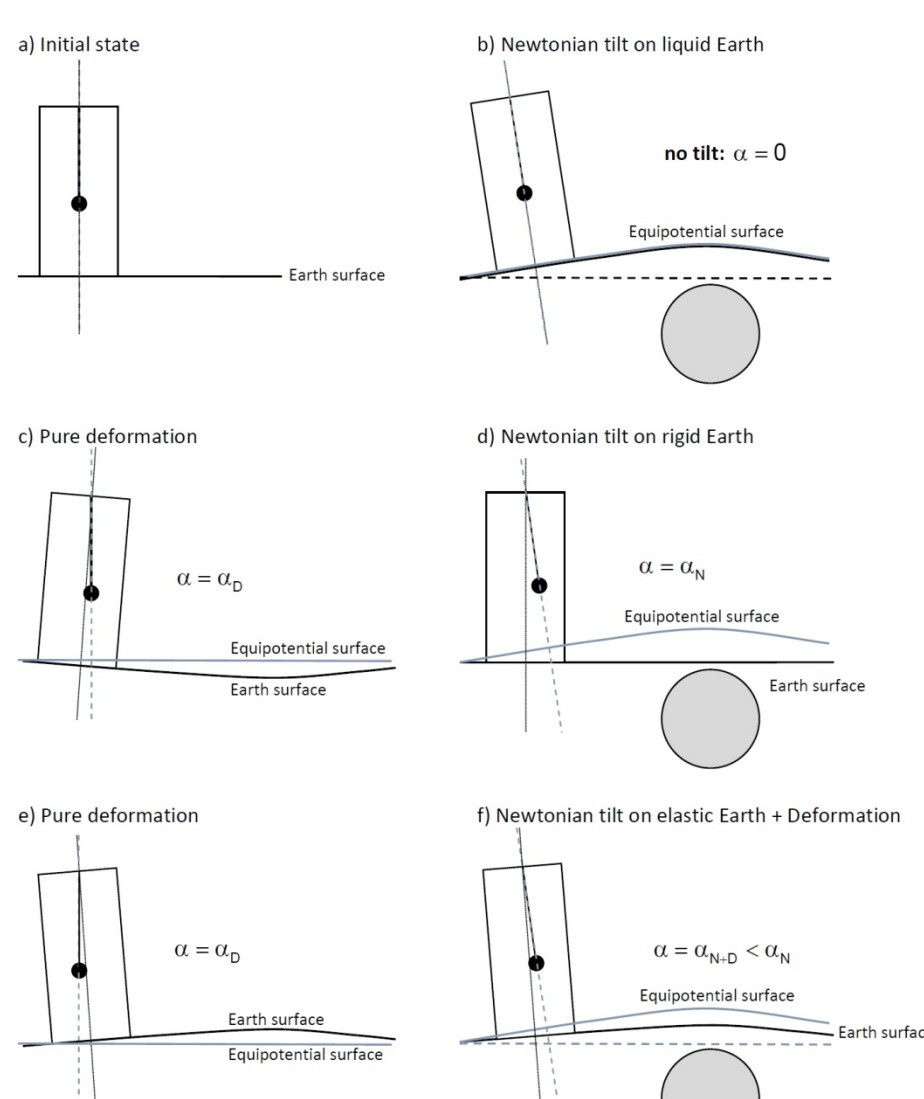

**Figure 1: Gravitational (Newtonian) tilt and deformation. Sphere: surplus mass (interior or exterior), equipotential surface (blue solid line), planet surface (solid black line), tilt sensor axis (dotted black line), plumb line (blue dashed line). a) Initial state, b) No tilt on a liquid planet, c) and e) Tilt due to deformation, d) Newtonian tilt on a rigid planet, f) Tilt on a deformable planet including both Newtonian tilt and tilt due to surface deformation.**

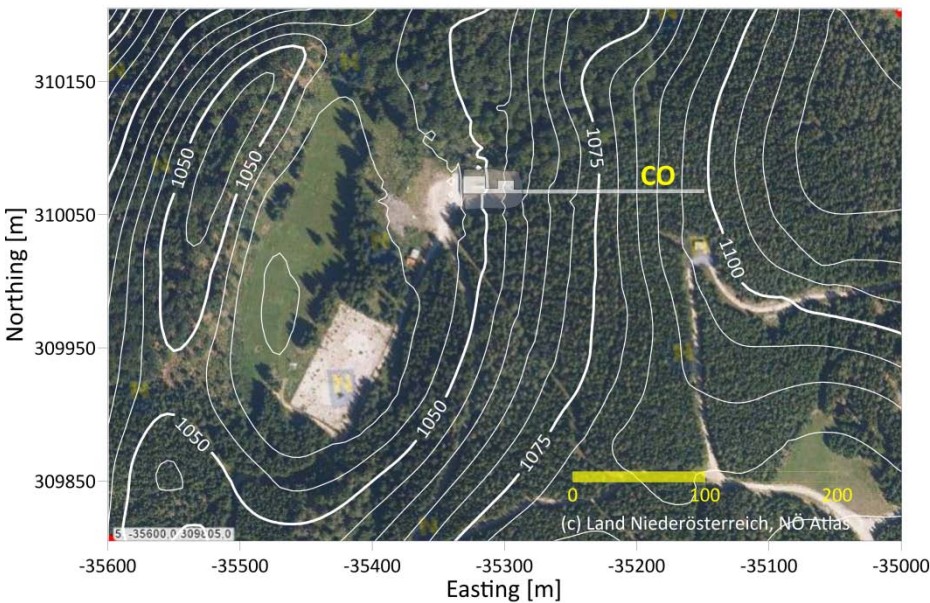

Figure 2: Map of the close surrounding of CO (©Land Niederösterreich, NÖ Atlas). Contour lines represent the topography elevation [m] of a high resolution DTM used for modelling. The outline of the observatory including the 150 m long tunnel is displayed as well. Details are presented in Fig. 3.

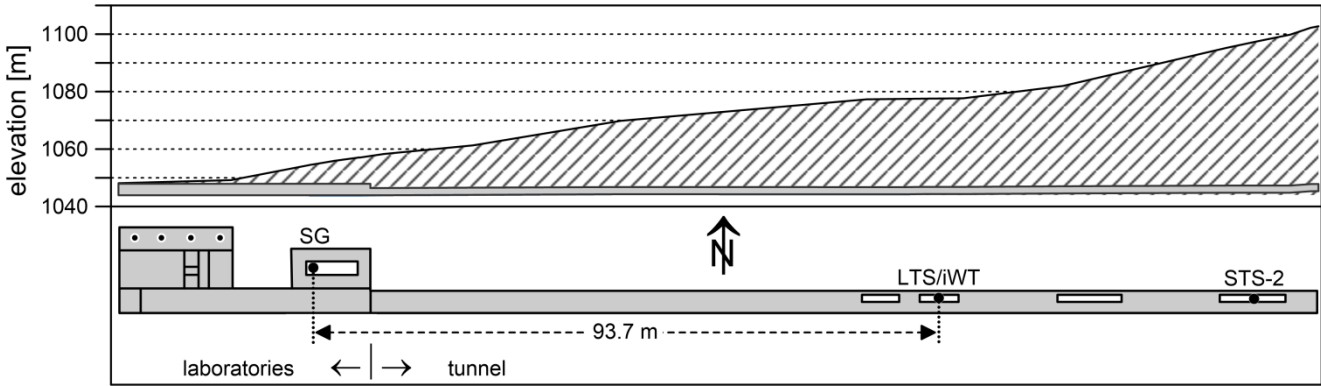

Figure 3: Vertical section and ground plan of the Conrad observatory. Sensor positions are displayed by black dots. Small dots indicate boreholes of different depth.

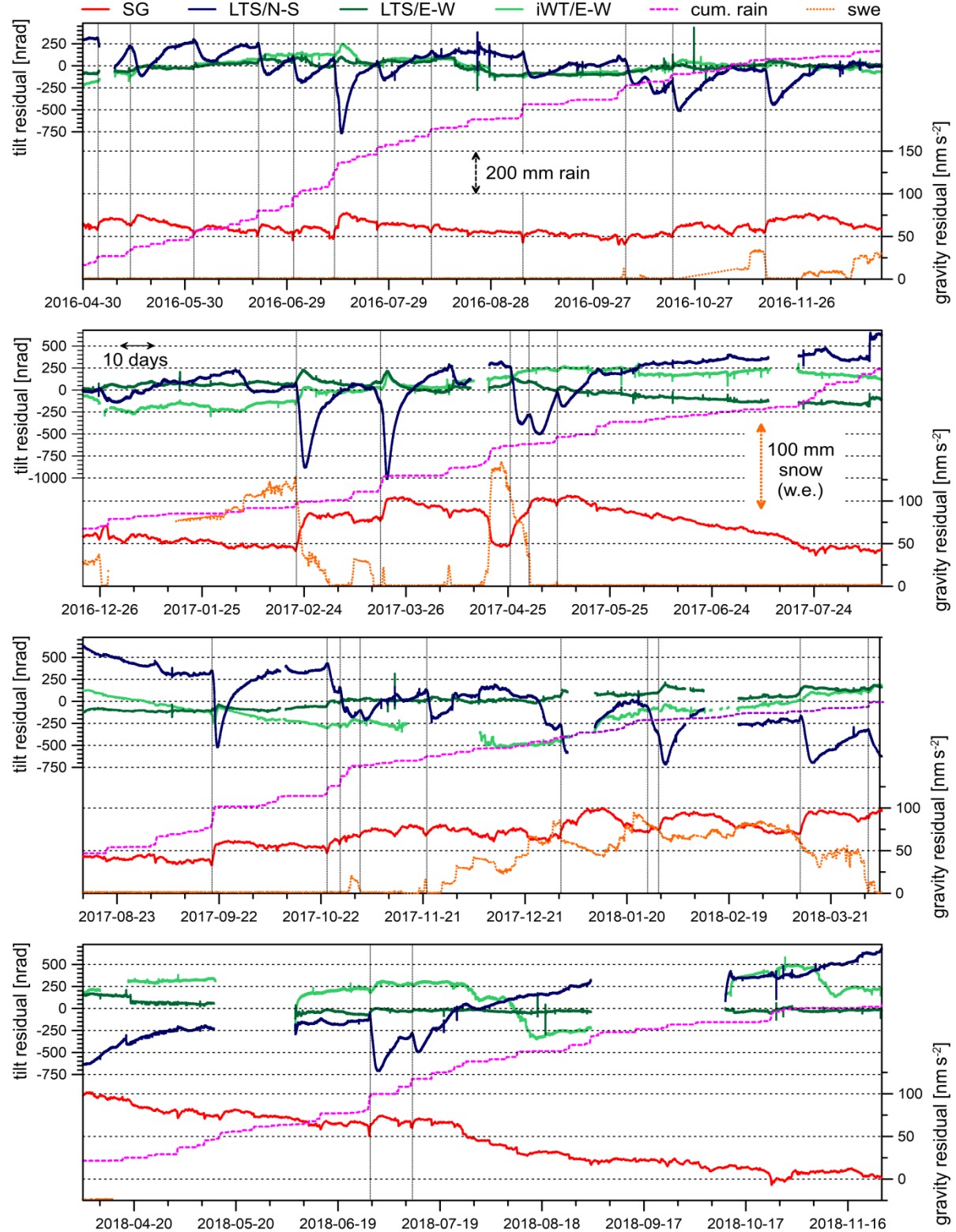

**Figure 4: Comparison of gravity and tilt residuals: gravity (red), N-S tilt (LTS, dark blue), E-W tilt (LTS: dark green, iWT: light**

green), cumulative rain (dashed magenta line); snow water equivalent (dotted orange line). Scales for rain and snow (water equivalent) are indicated by arrows. Vertical dotted lines mark the onset of hydrologically induced long-term events.

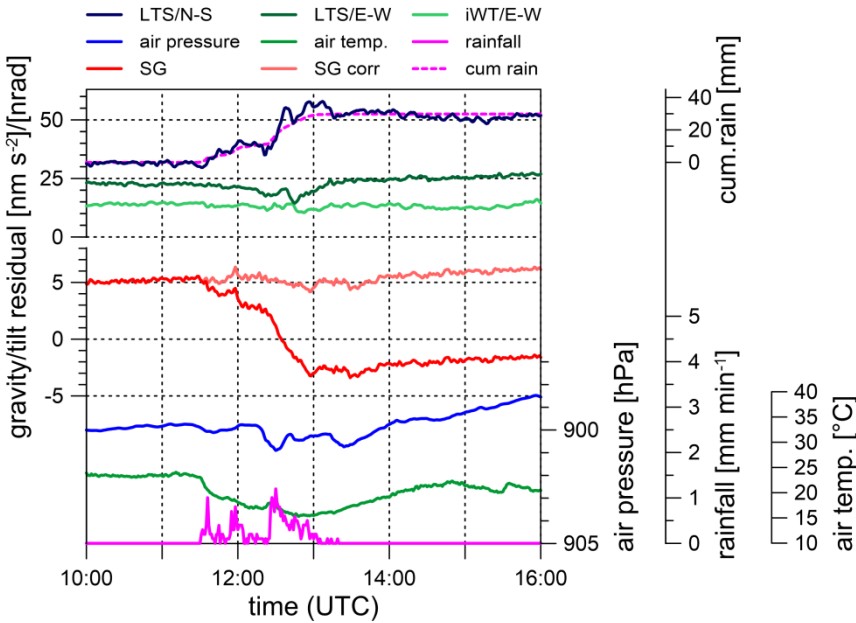

**Figure 5: Effect of heavy rain on gravity and tilt at CO on July 11, 2016. Gravity and N-S tilt residuals show patterns clearly**
**related to cumulative rain, while E-W tilts do not or only weakly respond to rain. Top: N-S tilt residuals (dark blue), E-W tilt residuals (LTS/dark green and iWT/light green), cumulative rain (dashed magenta line) scaled to fit the N-S tilt optimally. Middle: SG gravity residuals (red), gravity corrected for cumulative precipitation (light red). Bottom: rainfall (magenta), air pressure (blue) and outdoor air temperature (green).**

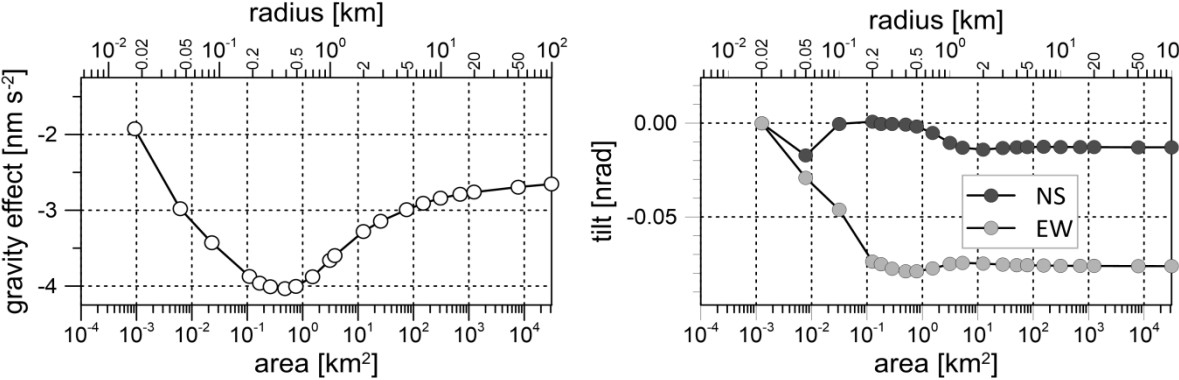

**Figure 6: Modelled gravitational effect of 10 mm rain on gravity (left) and tilt (right) at CO.**

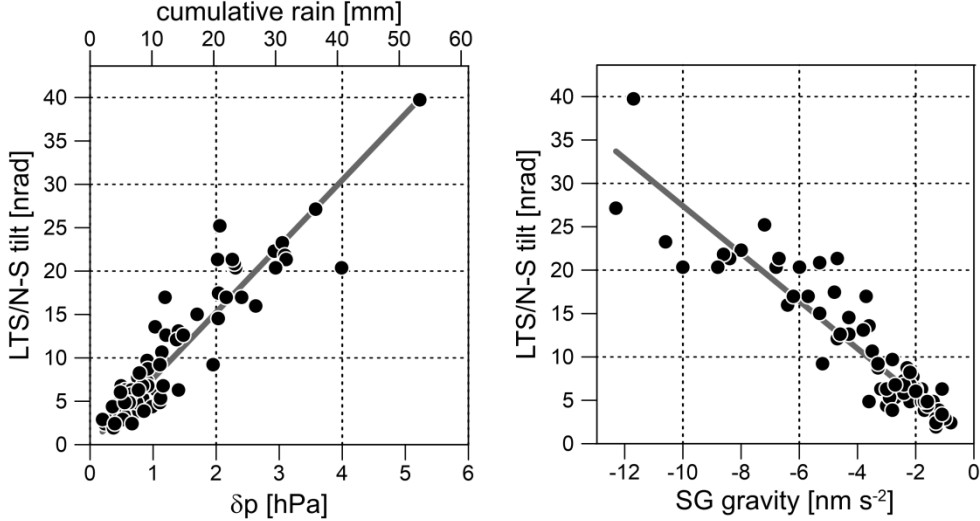

**Figure 7: Short-term N-S tilt and gravity residuals (water accumulation phase). N-S tilt response on cumulative rain at CO (left panel). Converting cumulative rain to surface pressure load reveals tilt to pressure admittance of 7.6 nrad hPa$^{-1}$ (solid line). Relation between gravity and N-S tilt residuals (right panel).**

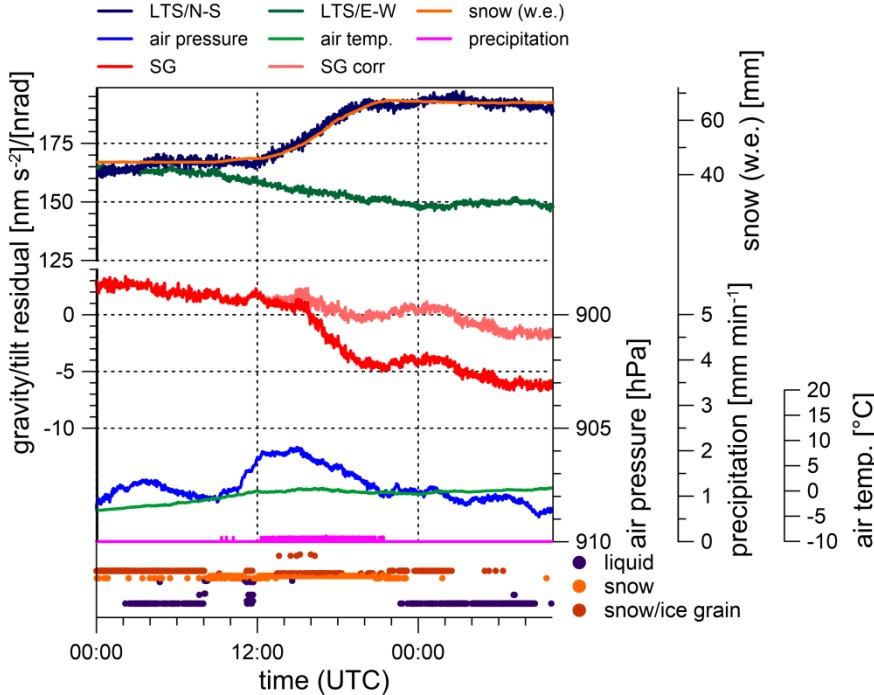

**Figure 8: Effect of snow accumulation on gravity and tilt at CO on December 21 and 22, 2017. Top: N-S tilt residuals (dark blue), E-W tilt residuals (dark green), snow (water equivalent, orange) scaled to fit the N-S tilt optimally. Middle: SG gravity residuals (red), gravity corrected for cumulative precipitation (light red). Bottom: air pressure (blue) and outdoor air temperature (green).**

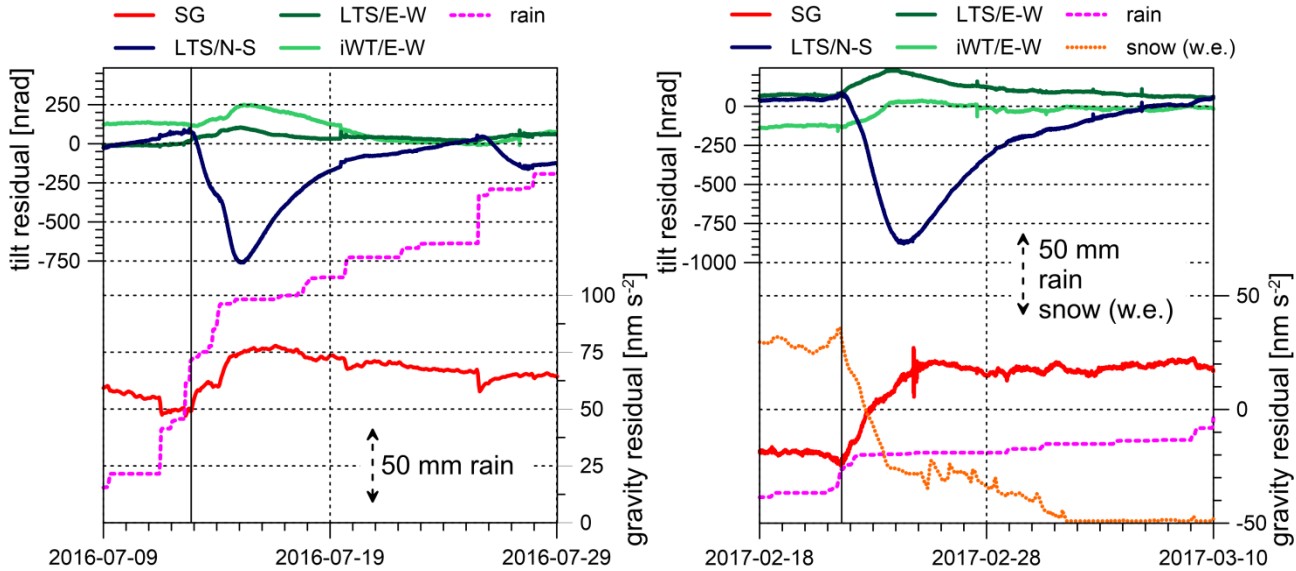

**Figure 9: Long-term gravity and tilt residual signals caused by hydrological processes after heavy and long-lasting rain (left panel) and during rapid snowmelt (right panel). Top: N-S tilt residuals (dark blue), E-W tilt residuals (LTS/dark green and iWT/light green). Bottom: SG gravity residuals (red). Cumulative rain (dashed magenta line), snow (water equivalent, dotted orange line); scales for rain and snow w.e. indicated by arrows. The black vertical line shows the onset of the long-term residual anomaly.**

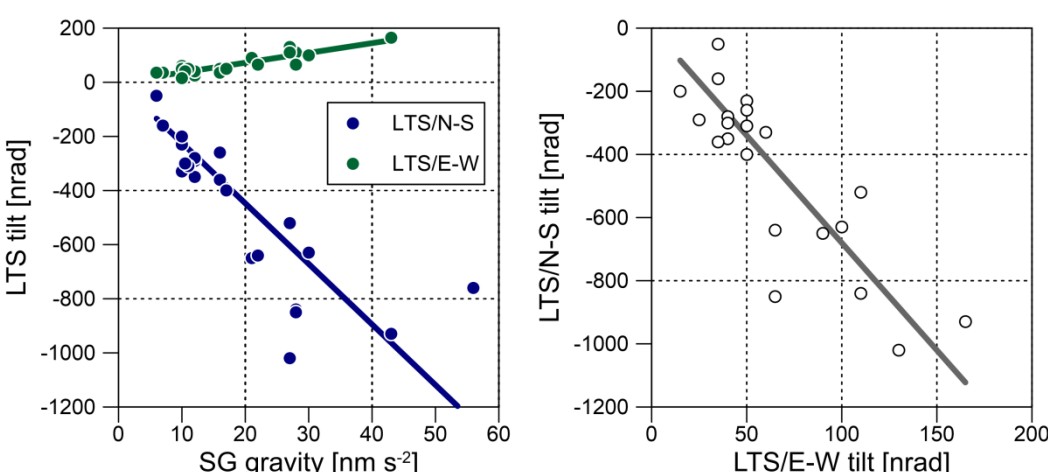

**Figure 10: Long-term tilt and gravity residuals (water percolation phase). Relation between the amplitudes of long-term tilt and gravity residual anomalies (left panel). Relation between the amplitudes of long-term LTS/N-S and LTS/E-W tilt residual anomalies (right panel). The average ratio of LTS/N-S to LTS/E-W tilt is −0.15.**

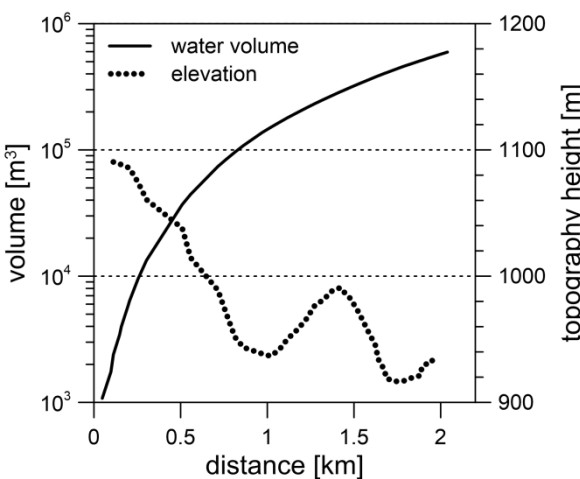

**Figure 11: Estimation of the magnitude of water volume (black) capable to produce 1 nrad tilt if it was purely Newtonian. The dotted line shows the cross section of the topography in the specific azimuth defined by the E-W and N-S tilts detected during rainfall events.**

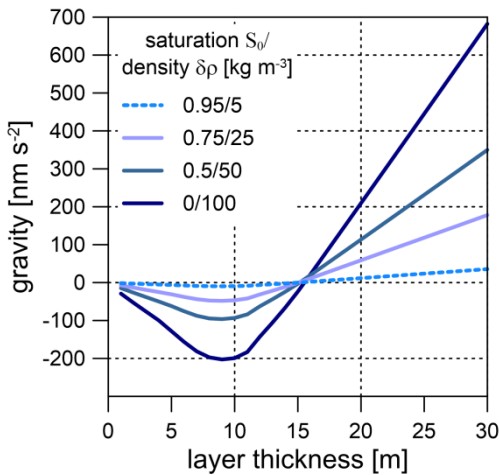

**Figure 12: Modelled gravity of a layer with constant thickness as function of the layer thickness and the degree of initial soil/rock saturation (for explanation see text).**

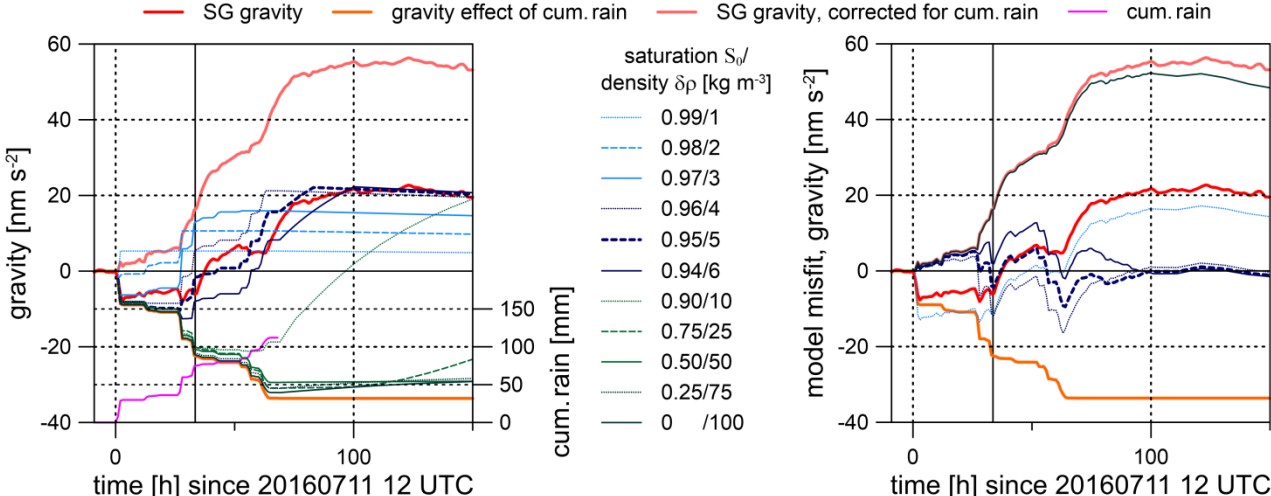

**Figure 13: Modelled gravity response for a real cumulative rain function (monitored between July 11 and July 14, 2016) for different degree of initial saturation $S_0$: Model response (left panel) and model mismatch (right panel). Observed gravity (red), cumulative rain (magenta) and gravity effect of cumulative rain (orange) are shown for comparison. Black vertical lines indicate the onset of the long-term anomaly. The right panel displays results only for the best fitting models ($0.94 \leq S_0 \leq 0.96$) and for $S_0 = 0$ and $S_0 = 0.99$.**

| Wave group | Tide model (DDW) | N-S | | | E-W | | | | | $\frac{\gamma(LTS)}{\gamma(iWT)}$ |
|---|---|---|---|---|---|---|---|---|---|---|
| | | LTS | | | | LTS | | iWT | | |
| Darwin-symbol | $\gamma$ | $amp_{theor}$ [nrad] | $\gamma$ $\sigma(\gamma)$ | $\varphi[°]$ $\sigma(\varphi)$ | $amp_{theor}$ [nrad] | $\gamma$ $\sigma(\gamma)$ | $\varphi[°]$ $\sigma(\varphi)$ | $\gamma$ $\sigma(\gamma)$ | $\varphi[°]$ $\sigma(\varphi)$ | |
| O1 | 0.6976 | 3.1275 | 1.0997 ±0.0358 | 7.235 ±1.867 | 23.4455 | 0.7135 ±0.0034 | −6.902 ±0.272 | 0.6746 ±0.0058 | −11.349 ±0.493 | 1.0577 |
| K1 | 0.7379 | 4.3962 | 1.2920 ±0.0259 | −1.318 ±1.152 | 32.9605 | 0.7773 ±0.0026 | −7.615 ±0.187 | 0.7278 ±0.0040 | −11.244 ±0.314 | 1.0680 |
| N2 | 0.6945 | 7.3013 | 0.6648 ±0.0067 | −2.691 ±0.579 | 9.8359 | 0.7673 ±0.0039 | −2.495 ±0.288 | 0.7130 ±0.0048 | −4.449 ±0.382 | 1.0762 |
| M2 | 0.6945 | 38.1330 | 0.6628 ±0.0013 | −3.193 ±0.115 | 51.3708 | 0.7401 ±0.0008 | −4.102 ±0.060 | 0.6849 ±0.0010 | −5.525 ±0.080 | 1.0806 |
| S2 | 0.6945 | 17.7398 | 0.6777 ±0.0031 | −2.880 ±0.264 | 23.8984 | 0.6896 ±0.0018 | −2.676 ±0.146 | 0.6203 ±0.0023 | −5.015 ±0.204 | 1.1117 |
| K2 | 0.6945 | 4.8195 | 0.6612 ±0.0129 | −2.531 ±1.116 | 6.4926 | 0.6880 ±0.0073 | −2.577 ±0.608 | 0.6385 ±0.0097 | −4.875 ±0.871 | 1.0775 |

**Table 1: Comparison of tidal parameters derived from tilt time series at CO. Theoretical body tide model: Dehant et al. (1999).**

| Air pressure admittance [nrad hPa$^{-1}$] | | |
|---|---|---|
| N-S | E-W | |
| LTS | LTS | iWT |
| 4.247 | 0.097 | −0.475 |
| ±0.034 | ±0.019 | ±0.034 |

**Table 2: Air pressure admittances in the diurnal and semidiurnal frequency band for the tilt sensors derived from tidal analysis.**


**Figure A1: Air pressure admittance function of the LTS/N-S tilt sensor derived from different observation periods covering a few days to less than 3 weeks each. Admittance (left), phase (middle), coherence (right). Circles and lines with intensive colours show the admittance (left), phase (middle) and coherence (right) respectively, averaged over the time series within 4 intervals (beginning - May 2017; May 2017 - June 2018/LTS repair; June 2018 - September 2018/LTS repair; September 2018 - end).**


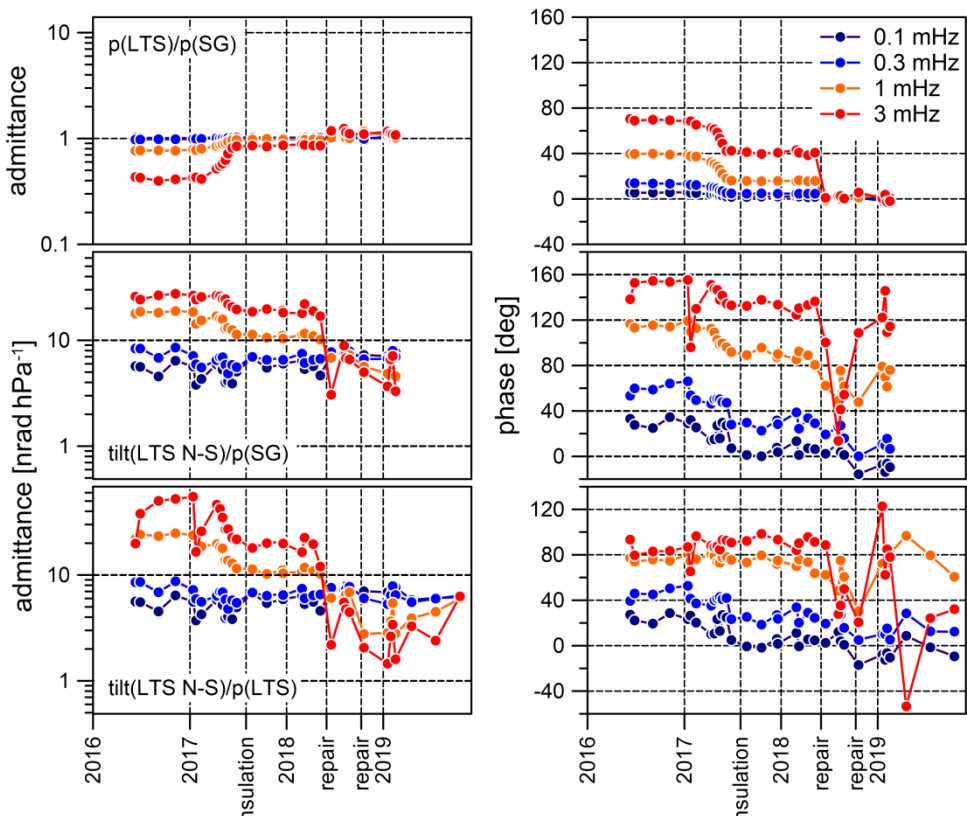

**Figure A2: Air pressure to tilt (N-S) admittance function and their temporal evolution at selected frequencies derived by using data from different air pressure sensors (bottom: LTS air pressure sensor; middle: SG air pressure sensor). The top panel shows the SG to LTS air pressure admittance function.**