# Peer review of "Hydrological Signals in Tilt and Gravity Residuals at Conrad Observatory (Austria)"

_Hydrology and Earth System Sciences, 2020_

## Referee Comment (RC1) · Anonymous Referee #1 · 13 Jul 2020

Review of "Hydrological Signals in tilt and gravity residuals at Conrad Observatory, Austria, by Meurers et al. This paper investigates the effects of precipitations on gravity and tilt recordings. Although precipitations are now well known (but difficult to model) effects, their influence on tilts or strains remains a poorly unexplored field, which deserves attention, especially in karst areas. This paper would deserve publication in HESS, but suffers presently from shortcomings and requires a major revision. General comments: Before providing detailed comments, I provide here the two most important points:

1. This paper provides a too detailed description of the processing of gravity and especially, tilt data. HESS is dedicated to hydro(geo)logy, hence, all the details of the SG processing are pure routine, the authors should just provide the basics and refer

to previous works, while I'm not sure that the whole discussion on the atmospheric pressure admittance of the tiltmeters is relevant. The authors could publish technical challenges into a more technical journal.

2. The discussion of the observed hydrogeological effects deserves more details. An important point is a decrease in gravity during precipitations that is immediately followed by an increase in gravity. If I see correctly, on 2016-07-11, Fig 4 shows the decrease, which is rather small comparing to the increase following just after, as shown by Figure 3 (same e.g. just before 2017-09-22 Fig 3). My interpretation is that during rainfall, gravity decreases because water is stored just above the gravimeter, and after while water percolates under the instrument (see similar effects in Watlet et al., WRR 2020). Of course, the response to rainfall probably depends on the degree of saturation of the saturated/unsaturated zones. Even if the authors do not dispose of groundwater measurements, they could build a simplistic model, to estimate the degree of saturation. A nice, original piece of information is provided by the tiltmeters: do they react more to the water stored immediately above the station, or more when water is supposedly underneath the gallery? Do you see a similar effect with snow, or not? As far as I can see, the snow does not affect tilts, hence we can rule Newtonian or load effects that would deform the rock around the gallery. Can you better quantify the effects of rain on gravity and tilts? Of course, there are admittances, but it is a general rule? Are there events obeying more the rule than others do? Are the responses of tilts and gravimeter perfectly proportional? So, elaborate, please.

Detailed comments

L10 An SG monitors changes in gravity

L11: add a blank: 5.5 m

L14: You should already mention the cavity effect, this is (unfortunately) an important effect

[Figure]

L20: unclear: what is exactly the difference between Newtonian and loading effects on tilts?

L25: in»at all spatial...

L27: loading: provide references

L30: add volcanoes

L34: You should mention the pioneering papers of Baker & Lennon and King & Bilham, both in Nature, 1973 (same remark on L179)

L41: complex infiltration process: mention that Conrad is a karstic area, where everything is expected to be even more complicated than in other hydrogeological contexts.

Section 2: provide a topographic map around the Conrad Observatory, showing the tunnel.

L48: I'd say: "Trafelberg at an elevation of 1050 m."

L53: there is no indication of the karstification, like e.g., sinkholes easy to detect?

L59: refer also to Van Camp, Meurers et al., J Geod 2016.

L63: What is "long one end"? Elaborate.

L71: in»at one end...

L72: 0.7x250: where does this '0.7' come from? "5.5 m/2 base": what does the "2" mean?

L73-74: "and an example can be given": strange sentence. Anyway, in my opinion, this belongs to useless technical details. In this paper, you should just mention than thermal effects are negligible (and this is especially true during rainfall, lasting only a few days in the worst case, while thermal effects would play a role only at longer periods or during maintenance).

L78-80: I do not understand the message. What is the relationship between the 50-100 m length and the resonances?

L103: "Thies": use the same wording as for Anton Paar: "A disdrometer (Thies). . .."

L109: nearby: provide an actual distance.

Section 3: too detailed, esp. for gravity.

L157: pole»polar

Section 4: the discussion on pressure is too detailed and somehow confusing.

L170: 5% of the tidal signal: do you mean that the observations are within 5% of the model? In that case, does the cavity effect play a role? Clarify, please.

L170: "much less data": provide the actual duration of both SG and tilt series.

L176: "However": what's exactly the link between the one-century old paper of Michelson, and the ocean loading at CO?

L181: calibrations errors: the tiltmeters are not at the same place, and have different baselines. Hence, they (probably?) undergo quite different cavity effects, and therefore, this may explain the differences, is it? Can you discuss this?

L194: sensor box: of LTS?

L194-203: this paragraph is not very clear and again, what is the relevant information for this study?

L213: temperature increase: I suppose that "temperature change" is more appropriate.

L214: the faster. . . faster": I do not understand. What is the message?

L219: why do we observe differences between the SG and tilt barometers? Different transfer functions? Also, why not directly comparing the barometers rather than working on the admittances?

L221: no idea about the steady change?

Section 5: could please better explain the Newtonian effect on tilts? Is it just due to a mass attracting more a side of the tiltmeter than the other side? It would be nice, perhaps in the introduction, to explain the different causes of tilts: Newtonian, loading (causing the crust to tilt), and infiltration in fissure and changes in pore pressure, and so on.

L261: the weak air pressure admittance...I do not understand your point. And, why is this admittance weaker than NS?

L265: looking at Figure 8 I see blue dots: it means rain, esp. before 12h or after 23h: could you explain, please?

L268-269: use UTC, avoid am and pm

L281: see also Watlet et al, WRR 2020.

L283-284: rather than charge and discharge I'd use "degree of saturation"

L285: do you mean the gravel layer above the concrete ceiling of the gallery? Unclear.

L288: Eventually: do you mean "perhaps"?

L293-294: "in advance": looking at the figure it's not so clear Could you quantify (e.g. by computing moving correlation)?

L308-310: I do not see your point: what's the relevancy of this information?

L314: gravity effects: but, your calculation of the rainfall admittance shows that you (nearly) perfectly model the Newtonian effects on gravity; unclear sentence.

L323: scale»scales

L329-335: you may quote Tenze et al., Bollettino di Geofisica Teorica ed Applicata, 2012

L330: which array?

L348-349: I do not understand: what is the link between the physics, hydrology and the cavity effect?

Figure 1: specify units on the vertical axis.

Figure 6: specify: modelled gravitational effect.

---

## Referee Comment (RC2) · Anonymous Referee #2 · 7 Sep 2020

This study reports on the effects of precipitation and snowmelt events on the recordings of a gravimeter and of tiltmeters that are located in an underground observatory. In both instrument types, signals related to these events can be recognized with different amplitudes and evolution in time. With this comparative analysis, the study makes a potentially valuable contribution to HESS in illustrating how geodetic monitoring methods might be of use for unraveling hydrological processes and water storage dynamics. However, in this perspective and to make the manuscript more accessible to the hydrological community, I suggest a revision of the manuscripts in particular with respect to the following:

In its present form, the manuscript does not make sufficiently clear how environmental processes such as variations in hydrological state variables (water storage) or water

fluxes may translate into the observation of the monitoring devices used here, i.e., gravimeters and tiltmeters. Given that the hydrological community is hardly familiar with gravimeters, and even less with tiltmeters, large part of the interpretation of the monitoring data presented in this study remains unclear or inconclusive to the reader as the basic idea behind these instruments is not sufficiently laid out.

Thus, I suggest to include in a revised version of the manuscript an introductory part that illustrates the measurement principle of gravimeter and tiltmeters and the influencing factors, and sets up general hypothesis how hydrological dynamics might be seen by these instruments, probably also highlighting in which way the instruments react differently to the same process. On these grounds, in the results and discussion chapters of the manuscript, explaining and discussing the observations at the Conrad Observatory can then be more clearly presented in particular with respect to the following issues:

- "Gravity and tilt residuals are associated to the same hydrological process but have different physical causes." (Abstract, ). What exactly are the physical causes that makes the difference between the instruments if the fundamental hydrological process is the same?

- The "cavity effect" is mentioned in several instances throughout the manuscript as an influencing factor (abstract, line 177, line 229, line 262) but it is not further explained. What is it about and how can it influence the observations? How can thus the statement "Because the tunnel axis is oriented in E-W direction, the N-S component corresponds to the tilt perpendicular to the tunnel axis and therefore is extremely sensitive to cavity effects." (line 177) be explained?

- Additional explanations to the two points before may also shed more light on ". . .because tilt is affected by the topography and by geometry and size of the cavity where the tilt meters are installed" (line 33). This sentence is not intelligible by its own for someone who is not familiar with tiltmeters.

- Line 227: "However, at long periods the air pressure signal in the tilt meter time series is due to geophysical/geodynamical reasons which are probably dominated by deformation due to air pressure loading." Also the gravimeter should be sensitive to loading effects that are associated with vertical displacements, right? Can this be jointly analyzed? More basically, even the term 'loading' might need to be explained for a hydrological reader.

- "Newtonian acceleration" Newtonian effect" (line 230, line 241) on the gravimeter needs to be explained with respect to water storage (mass) variations. Also, what is the difference to the 'Newtonian tilt effect' (line 243) seen by tiltmeters?

- Line 257: ". . . the observed total N-S tilt offsets as function of cumulative rain or the surface pressure load exerted by cumulative rain at the end of the respective rain event." Does a spatially uniform rain event cause a tilt signal? Probably not because also the surface pressure load is uniform? Thus, a tilt signal indicates spatially non-uniform rainfall?

- Line 262: "The short-term N-S tilt response is therefore interpretable as pure deformation effect (strain induced tilt) due to surface load, which is probably enhanced by the cavity effect." What does strain-induced tilt mean? How does this relate to the "cavity effect"?

- Line 274: "Therefore, deformation due to surface loading rather than due to pore pressure changes explains the observed short-term tilt signal." This statement is not clear a another effect is introduced that has not been explained before: how and why due pore pressure changes cause tilt signals? How do pore pressure changes relate to water storage changes that occur during a rainfall event?

- Line 298: ". . . a clear systematic tendency of the source azimuth (340° to 350°) is indicated." What does this mean? Needs some general introduction or explanation.

- Line 347: "It is not the physical source, but the hydrological process, which links the

residual anomalies of gravity and tilt." This statement is not clear. Is a hydrological process different from a physical source? What does this imply?

Other comments:

- Data section 2: I assume that there are no soil moisture or groundwater level data available at the observatory site or close to it? Is there a nearby river gauging station (or a smaller creek gauge) of which the discharge data could be used for comparing to the overall hydrological response of the study area?

- The manuscript ends rather abruptly. I suggest adding a concluding paragraph on what has been learned from this combined setup of gravimeters and tiltmeters towards their potential for unraveling water storage dynamics and hydrological processes, what are the limitations, what are additional observations that may be needed to disentangle ambiguities in these observations, or similar aspects.

---

## Editor Comment (EC1) · Marnik Vanclooster (Editor) · 9 Sep 2020

Dear Authors, We received 2 critical review reports on your manuscript. Both reviewers recognize the novelty of the study and recognize the potential of gravimeter and tilt measurements for unravelling hydrological processes. Yet both reviewers raise major concerns on the readibility of the manuscript, in particular for a hydrological audiance. Both reviewers also made suggestions to shape the manuscript so that it becomes aligned with the profile of HESS readership. Both reviewers also made detailed suggestions on more specific aspects. I concurr with these review reports and suggest you to proceed with your replies and major revision of your manuscript. The revised manuscript will be send out for review again to the original reviewers. Sincerely yours,

---

## Author Comment (AC1) · 7 Oct 2020

The authors thank Anonymous Referee #1 for the review with very useful comments and suggestions that certainly will improve the paper. We reply in the following and indicate how we plan to react in a revised version provided the editor decides accordingly. We add a reply to each single comment/suggestion of the reviewer.

1. "This paper provides a too detailed description of the processing of gravity and especially, tilt data. HESS is dedicated to hydro(geo)logy, hence, all the details of the SG processing are pure routine, the authors should just provide the basics and refer to previous works, while I'm not sure that the whole discussion on the atmospheric pressure admittance of the tiltmeters is relevant. The authors could publish technical

challenges into a more technical journal."

Reply: We agree with the reviewer that this part can be drastically shortened. However, we still consider the discussion of atmospheric pressure admittances to be necessary. We show in the paper, that short-term tilt residual anomalies associated instantaneously with heavy rain events exhibit on average a similar admittance as atmospheric pressure does at higher frequencies (about 0.3 mHz). From this observation we conclude that the short-term anomalies caused by rain are due to surface loading by accumulated rain water. At this point, we have to address the problem of admittance function changes at frequencies beyond 1 mHz caused by maintenance. We agree that presenting the details disrupts the flow of the text and we shift all the details into an appendix A.

2. "The discussion of the observed hydrogeological effects deserves more details. An important point is a decrease in gravity during precipitations that is immediately followed by an increase in gravity. If I see correctly, on 2016-07-11, Fig 4 shows the decrease, which is rather small comparing to the increase following just after, as shown by Figure 3 (same e.g. just before 2017-09-22 Fig 3). My interpretation is that during rainfall, gravity decreases because water is stored just above the gravimeter, and after while water percolates under the instrument (see similar effects in Watlet et al., WRR 2020). Of course, the response to rainfall probably depends on the degree of saturation of the saturated/unsaturated zones. Even if the authors do not dispose of groundwater measurements, they could build a simplistic model, to estimate the degree of saturation. A nice, original piece of information is provided by the tiltmeters: do they react more to the water stored immediately above the station, or more when water is supposedly underneath the gallery? Do you see a similar effect with snow, or not? As far as I can see, the snow does not affect tilts, hence we can rule Newtonian or load effects that would deform the rock around the gallery. Can you better quantify the effects of rain on gravity and tilts? Of course, there are admittances, but it is a general rule? Are there events obeying more the rule than others do? Are the responses of

tilts and gravimeter perfectly proportional? So, elaborate, please."

Reply: The event on 2016-07-11 is discussed in detail by Figs. 5 and 9 (please note, figure captions are always below the figure they refer to) discriminating between what we call "short-term" and "long-term" residual anomalies. Fig. 5 shows the instantaneous reaction of both gravity (decrease as water is above the sensor) and N-S tilt (increase due to deformation by surface load) to sudden rain accumulation. Actually, this behavior is seen in almost all (71 of 74) heavy rain events and in also case of fast snow accumulation (example provided in Fig. 8), i.e. snow does affect tilts similar as in case of rain. The associated signals are visible in Fig. 9 too, even if small compared to the long-term signal. The long-term anomalies start much later after sufficient water intrusion downwards into the subsurface has taken place (saturation). This process starts also in case of rapid snow melt as the example presented in Fig. 9 (right panel) shows. Short-term residual anomalies are quantified by Fig. 7. We will make this clearer. Quantifying the long-term anomalies is not easy because the tilt/gravity response to long-term water accumulation depends on the overall subsurface saturation for which we have no constraints based on observations. Nevertheless we will enhance the discussion and add some model calculations based on a simplistic models, which match the observations quite well. We also will address the relations between the gravity and tilt residuals quantitatively, both for the short-term and long-term residual anomalies.

L10 An SG monitors changes in gravity

Reply: Yes, provided the instrument axis is kept aligned to the plumb line. This is usually maintained by thermal levelers. If this system fails, the SG monitors the projection of the gravity vector onto the direction of the sensor axis. We agree: our formulation is not correct.

L11: add a blank: 5.5 m

L14: You should already mention the cavity effect, this is (unfortunately) an important

effect

Reply to L11 and L14: We agree and will change the text accordingly.

L20: unclear: what is exactly the difference between Newtonian and loading effects on tilts?

Reply: Please see our reply with respect to the general comment on section 5.

L25: in»at all spatial: : :

L27: loading: provide references

L30: add volcanoes

L34: You should mention the pioneering papers of Baker & Lennon and King & Bilham, both in Nature, 1973 (same remark on L179)

Reply to comments between L25 and L34: We agree and will consider the recommendations.

L41: complex infiltration process: mention that Conrad is a karstic area, where everything is expected to be even more complicated than in other hydrogeological contexts.

Reply: We agree and will add a corresponding statement already here.

Section 2: provide a topographic map around the Conrad Observatory, showing the tunnel.

Reply: We will add a topography map with the observatory outlines.

L48: I'd say: "Trafelberg at an elevation of 1050 m."

L53: there is no indication of the karstification, like e.g., sinkholes easy to detect?

Reply: There is a possibly a sinkhole filled with sediments 100-200 m apart from the observatory. Based on its geometry derived from geoelectric and seismic measurements the maximum estimate of the gravity effect of water accumulation is too small

for explain the observation. We add this information.

L59: refer also to Van Camp, Meurers et al., J Geod 2016.

Reply: We will add this reference.

L63: What is "long one end"? Elaborate.

L71: in»at one end: : :

L72: 0.7x250: where does this '0.7' come from? "5.5 m/2 base": what does the "2" mean?

L73-74: "and an example can be given": strange sentence. Anyway, in my opinion, this belongs to useless technical details. In this paper, you should just mention than thermal effects are negligible (and this is especially true during rainfall, lasting only a few days in the worst case, while thermal effects would play a role only at longer periods or during maintenance).

L78-80: I do not understand the message. What is the relationship between the 50-100 m length and the resonances?

Reply to comments related to L63-80: We agree that these technical details are not necessary and we will skip them.

L103: "Thies": use the same wording as for Anton Paar: "A disdrometer (Thies): : :."

Reply: We agree.

L109: nearby: provide an actual distance.

Reply: the distance is about 150 m. We will add this information.

Section 3: too detailed, esp. for gravity.

L157: pole»polar

Reply: We will shorten this chapter.

Section 4: the discussion on pressure is too detailed and somehow confusing.

Reply: We have re-organized this chapter shifting detailed information disrupting the flow of reading in Appendix A. We hope to have made the relevance of the air pressure admittance investigation for our study much clearer.

L170: 5% of the tidal signal: do you mean that the observations are within 5% of the model? In that case, does the cavity effect play a role? Clarify, please.

Reply: No. We refer to the RMS error of a single observation as derived from the LSQ adjustment of the tidal parameter (provided by ETERNA), taking M2 as reference.

L170: "much less data": provide the actual duration of both SG and tilt series.

Reply: SG time series covers 3512 days, LTS only 1064 days including gaps. Numbers of observations entering the adjustment are about 83500 (SG) and 21700 (LTS).

L176: "However": what's exactly the link between the one-century old paper of Michelson, and the ocean loading at CO?

Reply: We cancel this statement.

L181: calibrations errors: the tiltmeters are not at the same place, and have different baselines. Hence, they (probably?) undergo quite different cavity effects, and therefore, this may explain the differences, is it? Can you discuss this?

Reply: Base length is different, but both sensors monitor the tilt along the tunnel axis (E-W). Therefore we do not expect strong cavity effects for the E-W tilts. You are right, we cannot exclude that cavity effects play a role. We will formulate more carefully.

L194: sensor box: of LTS? Reply: Yes. We will make this clear.

L194-203: this paragraph is not very clear and again, what is the relevant information for this study?

Reply: Please refer to our reply to 1). Both atmospheric pressure and water accumulation due to rain or snow cause surface loading and deformation. We know that the air pressure admittance of the LTS has changed after maintenance work, but remained stable below 0.3 mHz. Hence, the admittances are comparable and not influenced by instrumental issues below 0.3 mHz.

L213: temperature increase: I suppose that "temperature change" is more appropriate.

Reply: We agree and change the wording.

L214: the faster::: faster": I do not understand. What is the message?

Reply: We know from the paper by Klügel (2003) that rapid air pressure changes can cause quasi-adiabatic temperature changes, which then cause tilt changes. We do not have temperature data in the tunnel accurate enough to reveal such small temperature changes. However, we have some hints that rapid air pressure variations are indeed associated with temperature changes even if we are not able to quantify. An air pressure pattern quickly passing the station will be seen as high frequency signature in the pressure time series; the faster the passing velocity the higher the frequency will appear.

L219: why do we observe differences between the SG and tilt barometers? Different transfer functions? Also, why not directly comparing the barometers rather than working on the admittances?

Reply: Yes, transfer functions differ. The LTS transfer function (tilt sensors as well as air pressure sensor) is not available so far. Determination would require interruption of the time series which is not convenient. In addition, now we know that the frequency transfer function is likely influenced by maintenance/transport which would require repeated in-situ determinations of the transfer function. Based on the atmospheric admittance investigations of the SG performed so far, we can assume the SG pressure sensor to be stable. That is why we take the SG pressure sensor as reference.

L221: no idea about the steady change?

Reply: Unfortunately not.

Section 5: could please better explain the Newtonian effect on tilts? Is it just due to a mass attracting more a side of the tiltmeter than the other side? It would be nice, perhaps in the introduction, to explain the different causes of tilts: Newtonian, loading (causing the crust to tilt), and infiltration in fissure and changes in pore pressure, and so on.

Reply: This is a very helpful suggestion. Newtonian tilt is the pure gravitational effect caused by any mass redistribution which changes the plumb line direction at the sensor location in case of a non-deformable planet. Loading (internal or external) effect on tilt is the tilt caused by deformation which changes the orientation of the surface the tilt sensor is mounted on. We will re-arrange the introduction considering the reviewer's suggestion.

L261: the weak air pressure admittance:::I do not understand your point. And, why is this admittance weaker than NS?

Reply: The tidal analysis shows that the air pressure admittance of the sensors sensitive to the E-W direction (along tunnel axis) is much smaller than for N-S. That means, surface load (either due to air pressure or rain/snow) does rarely produce clear signatures in the E-W tilts. We suppose this is due to the fact that the cavity effect is much smaller for E-W tilt than for N-S tilt. We will add explaining statements.

L265: looking at Figure 8 I see blue dots: it means rain, esp. before 12h or after 23h: could you explain, please?

Reply: Yes, this is correct. The information on the aggregate state of the precipitation particles is derived from the disdrometer data which is very sensitive to even tiny precipitation. As seen from the rain data (magenta color) no observable precipitation accumulation is observed, i.e. the liquid rain does not contribute essentially to water accumulation in the presented case study. We will make this clear in the text or figure

caption.

L268-269: use UTC, avoid am and pm

L281: see also Watlet et al, WRR 2020.

L283-284: rather than charge and discharge I'd use "degree of saturation"

Reply: We will consider suggestions given in comments to L268-L284.

L285: do you mean the gravel layer above the concrete ceiling of the gallery? Unclear.

Reply: No, we refer to the gravel sheet below the concrete base plate of the observatory building in front of the tunnel. Before construction of the building, a large amount of rock has been excavated. The remaining cragged and rough rock surface has been leveled by a gravel sheet before constructing the base plate. We will clarify this.

L288: Eventually: do you mean "perhaps"? Reply: Yes.

L293-294: "in advance": looking at the figure it's not so clear Could you quantify (e.g. by computing moving correlation)?

Reply: Thanks for the hint. Indeed there is weak evidence supporting the related statements, therefore we will cancel them.

L308-310: I do not see your point: what's the relevancy of this information?#

Reply: We will make calculation and conclusion clearer.

L314: gravity effects: but, your calculation of the rainfall admittance shows that you (nearly) perfectly model the Newtonian effects on gravity; unclear sentence.

Reply: Here, we refer to the long-term residual anomalies. Model calculations considering the magnitude of the associated E-W, N-W tilt and gravity residuals show that no point source is able to explain the observed signals. We make this clear.

L323: scale»scales

L329-335: you may quote Tenze et al., Bollettino di Geofisica Teorica ed Applicata, 2012

L330: which array?

L348-349: I do not understand: what is the link between the physics, hydrology and the cavity effect?

Figure 1: specify units on the vertical axis.

Figure 6: specify: modelled gravitational effect.

Reply to above comments: Yes, we will do and make the text clearer.

---

## Author Comment (AC2) · 7 Oct 2020

The authors thank Anonymous Referee #2 for the review with very useful comments and suggestions that certainly will improve the paper. We reply in the following and indicate how we plan to react in a revised version provided the editor decides accordingly. We add a reply to each single comment/suggestion of the reviewer.

This study reports on the effects of precipitation and snowmelt events on the recordings of a gravimeter and of tiltmeters that are located in an underground observatory. In both instrument types, signals related to these events can be recognized with different amplitudes and evolution in time. With this comparative analysis, the study makes a potentially valuable contribution to HESS in illustrating how geodetic monitoring meth-

ods might be of use for unraveling hydrological processes and water storage dynamics. However, in this perspective and to make the manuscript more accessible to the hydrological community, I suggest a revision of the manuscripts in particular with respect to the following: In its present form, the manuscript does not make sufficiently clear how environmental processes such as variations in hydrological state variables (water storage) or water fluxes may translate into the observation of the monitoring devices used here, i.e., gravimeters and tiltmeters. Given that the hydrological community is hardly familiar with gravimeters, and even less with tiltmeters, large part of the interpretation of the monitoring data presented in this study remains unclear or inconclusive to the reader as the basic idea behind these instruments is not sufficiently laid out. Thus, I suggest to include in a revised version of the manuscript an introductory part that illustrates the measurement principle of gravimeter and tiltmeters and the influencing factors, and sets up general hypothesis how hydrological dynamics might be seen by these instruments, probably also highlighting in which way the instruments react differently to the same process. On these grounds, in the results and discussion chapters of the manuscript, explaining and discussing the observations at the Conrad Observatory can then be more clearly presented in particular with respect to the following issues: - "Gravity and tilt residuals are associated to the same hydrological process but have different physical causes." (Abstract, ). What exactly are the physical causes that makes the difference between the instruments if the fundamental hydrological process is the same?

Reply: Thanks for this valuable suggestion. We will completely re-organize the structure of the paper and add a detailed text explaining the different sensitivity to physical processes like gravitation and deformation and what the sensors are able to detect. We also explain disturbing effects particularly for the tiltmeters (cavity effect) and why this effect is stronger for the N-S tilt component. References to published results based on using gravimeters and tiltmeters in hydro-geological studies will be provided in the introduction already. We will omit technical details or shift them to an appendix for interested readers.

- The "cavity effect" is mentioned in several instances throughout the manuscript as an influencing factor (abstract, line 177, line 229, line 262) but it is not further explained. What is it about and how can it influence the observations? How can thus the statement "Because the tunnel axis is oriented in E-W direction, the N-S component corresponds to the tilt perpendicular to the tunnel axis and therefore is extremely sensitive to cavity effects." (line 177) be explained?

- Additional explanations to the two points before may also shed more light on ": : :because tilt is affected by the topography and by geometry and size of the cavity where the tilt meters are installed" (line 33). This sentence is not intelligible by its own for someone who is not familiar with tiltmeters.

Reply to both comments: Re-organizing our paper will give opportunity to address the problem already in the introduction and to make it clear (please refer to our reply above).

- Line 227: "However, at long periods the air pressure signal in the tilt meter time series is due to geophysical/geodynamical reasons which are probably dominated by deformation due to air pressure loading." Also the gravimeter should be sensitive to loading effects that are associated with vertical displacements, right? Can this be jointly analyzed? More basically, even the term 'loading' might need to be explained for a hydrological reader.

Reply: Generally a gravimeter is sensitive to deformation effects as well: if deformation is associated with vertical displacement of the Earth's surface, then the gravimeter moves within the gravity field of the Earth and experiences different gravity. Generally upward and downward movement decreases and increases gravity, respectively. Secondly, deformation always means mass redistribution and consequently a gravity change. However, at local spatial scale, vertical displacement generated by air pressure variation is very small and the associated gravity effect is negligible. At regional or global scale such effects have to be considered. Another effect of the displacement

is inertial acceleration if the displacement is time dependant (seismometer principle). However, this is important only for high frequencies (> 0.1 mHz). We will explain this briefly in a next version of our paper.

- "Newtonian acceleration" Newtonian effect" (line 230, line 241) on the gravimeter needs to be explained with respect to water storage (mass) variations. Also, what is the difference to the 'Newtonian tilt effect' (line 243) seen by tiltmeters?

Reply: "Newtonian" is another wording meaning "gravitational". Any mass movement in the atmosphere and hydrosphere (air mass, water etc.) changes the gravitational potential and consequently the gravity field. Gravimeters are sensitive to the vertical component (or the norm) of the gravity vector while tiltmeters are sensitive to the horizontal component. We will explain this briefly in the introduction.

- Line 257: ": : : the observed total N-S tilt offsets as function of cumulative rain or the surface pressure load exerted by cumulative rain at the end of the respective rain event." Does a spatially uniform rain event cause a tilt signal? Probably not because also the surface pressure load is uniform? Thus, a tilt signal indicates spatially nonuniform rainfall?

Reply: The gravitational effect of rain/snow water is too small to emerge out of the noise of the tiltmeters. However, like atmospheric pressure variations, also water at the surface exerts pressure onto the surface which results to deformation. Tiltmeters are sensitive to even little deformation and therefore experience a clear signal. This signal might be different if the load is uniform or non-uniform, but this cannot be addressed without further investigation and is out of the scope of our paper.

- Line 262: "The short-term N-S tilt response is therefore interpretable as pure deformation effect (strain induced tilt) due to surface load, which is probably enhanced by the cavity effect." What does strain-induced tilt mean? How does this relate to the "cavity effect"?

Reply: The cavity effect belongs to strain-induced tilt. You are right, our formulation is confusing, and we will correct this.

- Line 274: "Therefore, deformation due to surface loading rather than due to pore pressure changes explains the observed short-term tilt signal." This statement is not clear a another effect is introduced that has not been explained before: how and why due pore pressure changes cause tilt signals? How do pore pressure changes relate to water storage changes that occur during a rainfall event?

Reply: Water percolation into the subsurface or injection of water in boreholes is able to change the pore pressure which causes deformation. We will address this in the introduction.

- Line 298: ": : : a clear systematic tendency of the source azimuth (340_ to 350_) is indicated." What does this mean? Needs some general introduction or explanation.

Reply: Comparing the E-W and N-S tilt data, the amplitude ratio of the long-term residual anomalies turns out to be about $-0.15$ on average; E-W tilt is always positive, N-S tilt is always negative. If the observed tilt is caused by gravitational attraction by a volume of stored water, then the source must be located on a line with azimuth of about 170 (the azimuth of 340 to 350° mentioned in the submitted MS was a misprint). We will clarify this.

- Line 347: "It is not the physical source, but the hydrological process, which links the residual anomalies of gravity and tilt." This statement is not clear. Is a hydrological process different from a physical source? What does this imply?

Reply: The hydrological process is water accumulation (short-term residual anomalies) or infiltration (long-term anomalies). The physical reason for the reaction of gravimeters is different from that of tiltmeters. Gravity residuals predominantly reflect the gravitational effect of water mass accumulation close to the surface (short-term) or water transport downwards (long-term). In contrast, the tiltmeters reflect the tilt due to deformation by the water load on topography (short-term) or due to deformation by water percolated into the subsurface (long-term). We think all this gets clearer after having changed the introduction and the discussion accordingly.

Other comments: - Data section 2: I assume that there are no soil moisture or groundwater level data available at the observatory site or close to it? Is there a nearby river gauging station (or a smaller creek gauge) of which the discharge data could be used for comparing to the overall hydrological response of the study area?

Reply: You are right. Unfortunately we do not have any hydrological instrumentation. There is river at the foot of Trafelberg mountain, however it would be unclear what amount of water in the creek stems from the Trafelberg massif or from other catchment areas.

- The manuscript ends rather abruptly. I suggest adding a concluding paragraph on what has been learned from this combined setup of gravimeters and tiltmeters towards their potential for unraveling water storage dynamics and hydrological processes, what are the limitations, what are additional observations that may be needed to disentangle ambiguities in these observations, or similar aspects.

Reply: We will add a short paragraph on these aspects at the end.
* * *

---

## Author Response (AR1)

**Author's response**

The authors are very grateful to both referees for their careful review providing very helpful comments and suggestions that improved the paper. We first indicate our reaction to comments and suggestions by adding a reply to each single comment/suggestion of the reviewers and present then a list of relevant changes followed by a manuscript allowing for tracing the changes.

*Reviewer 1*:

1. "This paper provides a too detailed description of the processing of gravity and especially, tilt data. HESS is dedicated to hydro(geo)logy, hence, all the details of the SG processing are pure routine, the authors should just provide the basics and refer to previous works, while I'm not sure that the whole discussion on the atmospheric pressure admittance of the tiltmeters is relevant. The authors could publish technical challenges into a more technical journal."
Reply:
We agree with the reviewer that this part can be drastically shortened. However, we still consider the discussion of atmospheric pressure admittances to be necessary. We show in the paper, that short-term tilt residual anomalies associated instantaneously with heavy rain events exhibit a similar admittance as atmospheric pressure does at higher frequencies (about 0.3 mHz). From this observation we conclude that the short-term anomalies caused by rain are due to surface loading by accumulated rain water. At this point, we have to address the problem of admittance function changes at frequencies beyond 1 mHz caused by maintenance. We agree that presenting the details disrupts the flow of the text and we shifted all the details into an appendix A.

2. "The discussion of the observed hydrogeological effects deserves more details. An important point is a decrease in gravity during precipitations that is immediately followed by an increase in gravity. If I see correctly, on 2016-07-11, Fig 4 shows the decrease, which is rather small comparing to the increase following just after, as shown by Figure 3 (same e.g. just before 2017-09-22 Fig 3). My interpretation is that during rainfall, gravity decreases because water is stored just above the gravimeter, and after while water percolates under the instrument (see similar effects in Watlet et al., WRR 2020). Of course, the response to rainfall probably depends on the degree of saturation of the saturated/unsaturated zones. Even if the authors do not dispose of groundwater measurements, they could build a simplistic model, to estimate the degree of saturation. A nice, original piece of information is provided by the tiltmeters: do they react more to the water stored immediately above the station, or more when water is supposedly underneath the gallery? Do you see a similar effect with snow, or not? As far as I can see, the snow does not affect tilts, hence we can rule Newtonian or load effects that would deform the rock around the gallery. Can you better quantify the effects of rain on gravity and tilts? Of course, there are admittances, but it is a general rule? Are there events obeying more the rule than others do? Are the responses of tilts and gravimeter perfectly proportional? So, elaborate, please."
Reply:
Fig. 5 shows the instantaneous reaction of both gravity (decrease as water is above the sensor) and N-S tilt (increase due to deformation by surface load) to sudden rain accumulation. Actually, this behavior is seen in almost all (71 of 74) heavy rain events and in also case of fast snow accumulation (example provided in Fig. 8 (revised MS), i.e. snow does affect tilts similar as in case of rain. The associated signals are visible in Fig. 9 too, even if small compared to the long-term signal. The long-term anomalies start much later after sufficient water intrusion downwards into the subsurface has taken place (saturation). This process starts also in case of rapid snow melt as the example presented in Fig. 9 (right panel) shows. Short-term residual anomalies are quantified by Fig. 7. Quantifying the long-term anomalies is not easy because the tilt/gravity response to long-term water accumulation depends on the overall subsurface saturation for which we have no constraints based on observations. Nevertheless providing a simple model is a valuable suggestion we tried to follow. We added some model calculations based on a simplistic models, which match the observations quite well. We also addressed the relations between gravity and tilt residuals quantitatively, both for the short-term and long-term residual anomalies.

L10 An SG monitors changes in gravity
Reply:

Yes, provided the instrument axis is kept aligned to the plumb line. This is usually maintained by thermal levelers. If this system fails, the SG monitors the projection of the gravity vector onto the direction of the sensor axis. However with respect to modeling, we need to consider only the vertical component of disturbing gravitational vectors, because the horizontal ones are small compared to the magnitude of gravity. This was the background of our formulation in the abstract. We agree: our formulation is not correct and we changed it accordingly without going into details.

55

L11: add a blank: 5.5 m
L14: You should already mention the cavity effect, this is (unfortunately) an important effect
Reply to L11 and L14:
60 We changed the text accordingly.

L20: unclear: what is exactly the difference between Newtonian and loading effects on tilts?
Reply:
We have completely re-structured the introduction as also proposed by reviewer 2. Please see our reply with respect to the
65 general comment on section 5.

L25: in»at all spatial: : :
L27: loading: provide references
L30: add volcanoes
70 L34: You should mention the pioneering papers of Baker & Lennon and King & Bilham, both in Nature, 1973 (same remark on L179)
Reply to comments between L25 and L34:
We considered all these recommendations and give credit to Baker, King and Bilham.

75 L41: complex infiltration process: mention that Conrad is a karstic area, where everything is expected to be even more complicated than in other hydrogeological contexts.
Reply:
We mention the strain-tilt coupling now already in the abstract.

80 Section 2: provide a topographic map around the Conrad Observatory, showing the tunnel.
Reply:
We added a topography map showing the observatory outlines.

L48: I'd say: "Trafelberg at an elevation of 1050 m."
85 L53: there is no indication of the karstification, like e.g., sinkholes easy to detect?
Reply:
There is possibly a sinkhole filled with sediments 100-200 m apart from the observatory. Based on its geometry derived from geoelectric and seismic measurements the maximum estimate of the gravity effect of water accumulation is too small for explaining the observation. We added this information in the introduction.
90

L59: refer also to Van Camp, Meurers et al., J Geod 2016.
Reply:
We added this reference.

95 L63: What is "long one end"? Elaborate.
L71: in»at one end: : :
L72: 0.7x250: where does this '0.7' come from? "5.5 m/2 base": what does the "2" mean?
L73-74: "and an example can be given": strange sentence. Anyway, in my opinion, this belongs to useless technical details. In this paper, you should just mention than thermal effects are negligible (and this is especially true during rainfall, lasting
100 only a few days in the worst case, while thermal effects would play a role only at longer periods or during maintenance).

L78-80: I do not understand the message. What is the relationship between the 50-100 m length and the resonances?

Reply to comments related to L63-80:

Usually, water level tiltmeters observe at both ends of the tube; this type has only one interferometer at one end. We agree that these technical details are not necessary and we skipped them.

L103: "Thies": use the same wording as for Anton Paar: "A disdrometer (Thies): : :."

Reply:

We changed the wording.

L109: nearby: provide an actual distance.

Reply:

The distance is about 150 m. We added this information.

Section 3: too detailed, esp. for gravity.

L157: pole»polar

Reply:

We have removed the details and referred to literature, which shortened this chapter remarkably.

Section 4: the discussion on pressure is too detailed and somehow confusing.

Reply:

We have re-organized this chapter shifting detailed information disrupting the flow of reading to Appendix A. Also, we made the relevance of the air pressure admittance investigation for our study much clearer.

L170: 5% of the tidal signal: do you mean that the observations are within 5% of the model? In that case, does the cavity effect play a role? Clarify, please.

Reply:

No. We refer to the RMS error of a single observation as derived from the LSQ adjustment of the tidal parameter (provided by ETERNA), taking M2 as reference. We re-formulated accordingly.

L170: "much less data": provide the actual duration of both SG and tilt series.

Reply:

SG time series covers 3512 days, LTS only 1064 days including gaps. Numbers of observations entering the adjustment are about 83500 (SG) and 21700 (LTS). We provided these numbers.

L176: "However": what's exactly the link between the one-century old paper of Michelson, and the ocean loading at CO?

Reply:

We cancelled this statement.

L181: calibrations errors: the tiltmeters are not at the same place, and have different baselines. Hence, they (probably?) undergo quite different cavity effects, and therefore, this may explain the differences, is it? Can you discuss this?

Reply:

Base length is different, but both sensors monitor the tilt along the tunnel axis (E-W). Therefore we do not expect strong cavity effects for the E-W tilts. However, you are right, we cannot exclude that cavity effects play a role as well. We formulated more carefully.

L194: sensor box: of LTS?

Reply:

Yes. We made this clear.

L194-203: this paragraph is not very clear and again, what is the relevant information for this study?

Reply:

Please refer to our reply to 1). Both atmospheric pressure and water accumulation due to rain or snow cause deformation by surface loading. We know that the air pressure admittance of the LTS has changed after maintenance work, but remained stable below 0.3 mHz. Hence, the admittances are comparable and not influenced by instrumental issues below 0.3 mHz.

155

L213: temperature increase: I suppose that "temperature change" is more appropriate.
Reply:
We agree and changed the wording.

160 L214: the faster: : : faster": I do not understand. What is the message?
Reply:
We know from the paper by Klügel (2003) that rapid air pressure changes can cause quasi-adiabatic (isentropic) temperature changes, which then cause tilt changes. We do not have temperature data in the tunnel accurate enough to reveal such small temperature changes. However, we have some hints that rapid air pressure variations are indeed associated with temperature
165 changes even if we are not able to quantify. An air pressure pattern quickly passing the station will be seen as high frequency signature in the pressure time series; the faster the passing velocity the higher the frequency will appear.

L219: why do we observe differences between the SG and tilt barometers? Different transfer functions? Also, why not directly comparing the barometers rather than working on the admittances?
170 Reply:
Yes, transfer functions differ. The LTS transfer function (tilt sensors as well as air pressure sensor) is not available so far. Determination would require interruption of the time series which is not convenient. In addition, now we also know that the frequency transfer function is likely influenced by maintenance/transport which would require repeated in-situ determinations of the transfer function. Based on the atmospheric admittance investigations of the SG performed so far, we
175 can assume the SG pressure sensor to be stable. That is why we take the SG pressure sensor as reference.

L221: no idea about the steady change?
Reply:
Unfortunately not; we contacted the manufacturer, however, Erich Lippmann has no idea either.
180

Section 5: could please better explain the Newtonian effect on tilts? Is it just due to a mass attracting more a side of the tiltmeter than the other side? It would be nice, perhaps in the introduction, to explain the different causes of tilts: Newtonian, loading (causing the crust to tilt), and infiltration in fissure and changes in pore pressure, and so on.
Reply:
185 This is a very helpful suggestion. Newtonian tilt is the pure gravitational effect caused by any mass redistribution which changes the plumb line direction at the sensor location in case of a non-deformable planet. Loading (internal or external) effect on tilt is the tilt caused by deformation which changes the orientation of the surface the tilt sensor is mounted on. We re-arranged the introduction completely and added an illustrated explanation. We also refer to general and hydrology related investigation of tilt already in the introduction.
190

L261: the weak air pressure admittance: : :I do not understand your point. And, why is this admittance weaker than NS?
Reply:
The tidal analysis shows that the air pressure admittance of the sensors sensitive to the E-W direction (along tunnel axis) is much smaller than for N-S. This means that surface load (either due to air pressure or rain/snow) does rarely produce clear
195 signatures in the E-W tilts. We suppose this is due to the fact that the cavity effect is much smaller for E-W tilt than for N-S tilt. We added explaining statements.

L265: looking at Figure 8 I see blue dots: it means rain, esp. before 12h or after 23h: could you explain, please?
Reply:

200   Yes, this is correct. The information on the aggregate state of the precipitation particles is derived from the disdrometer data, which is very sensitive to even tiny precipitation. However, as shown by the rain data (magenta color) no observable precipitation accumulation is observed, i.e. the liquid rain does not essentially contribute to water accumulation in the presented case study. We made this clear in the text.

205   L268-269: use UTC, avoid am and pm
      L281: see also Watlet et al, WRR 2020.
      L283-284: rather than charge and discharge I'd use "degree of saturation"
      Reply:
      We considered all suggestions provided in comments to L268-L284.
210
      L285: do you mean the gravel layer above the concrete ceiling of the gallery? Unclear.
      Reply: No, we refer to the gravel sheet below the concrete foundation plate of the observatory building in front of the tunnel. Before construction of the building, a large amount of rock has been excavated. The remaining cragged and rough rock surface has been leveled by a gravel sheet before constructing the foundation plate. We clarified this in chapter 2.
215
      L288: Eventually: do you mean "perhaps"?
      Reply:
      Yes.

220   L293-294: "in advance": looking at the figure it's not so clear Could you quantify (e.g. by computing moving correlation)?
      Reply:
      Thanks for the hint. Actually, there is only weak evidence supporting the related statements, therefore we cancelled them.

      L308-310: I do not see your point: what's the relevancy of this information?
225   Reply:
      We checked, whether one single source can cause the observed long-term tilt residuals, assuming they were purely gravitational. We made the text clearer.

      L314: gravity effects: but, your calculation of the rainfall admittance shows that you (nearly) perfectly model the Newtonian
230   effects on gravity; unclear sentence.
      Reply:
      Yes, this holds for the short-term residual anomalies, i.e. the instantaneous response to water accumulation at the ground. Here, we refer to the long-term residual anomalies. Model calculations considering the magnitude of the associated E-W, N-W tilt and gravity residuals show that no point source is able to explain the observed signals. We made this clear.
235
      L323: scale»scales
      L329-335: you may quote Tenze et al., Bollettino di Geofisica Teorica ed Applicata, 2012
      L330: which array?
      Reply to comments L323-330:
240   We considered these suggestions.

      L348-349: I do not understand: what is the link between the physics, hydrology and the cavity effect?
      Reply:
      The hydrological process, either water accumulation (short-term) or subsurface infiltration (long-term) is the link between
245   gravity and tilt residuals. Gravity and tilt respond to these processes based on different physical phenomena: gravitational effects of the moving water mass (gravity) vs. deformation by loading. The cavity effect enhances the tilt component perpendicular to the tunnel axis due to strain-tilt coupling. We extended the text to make this clear.

      Figure 1: specify units on the vertical axis.

250     Figure 6: specify: modelled gravitational effect.
        Reply: We changed Figs. 1 and 6 accordingly.

        *Reviewer 2*:
255
        This study reports on the effects of precipitation and snowmelt events on the recordings of a gravimeter and of tiltmeters that
        are located in an underground observatory. In both instrument types, signals related to these events can be recognized with
        different amplitudes and evolution in time. With this comparative analysis, the study makes a potentially valuable
        contribution to HESS in illustrating how geodetic monitoring methods might be of use for unraveling hydrological processes
260     and water storage dynamics.
        However, in this perspective and to make the manuscript more accessible to the hydrological community, I suggest a
        revision of the manuscripts in particular with respect to the following: In its present form, the manuscript does not make
        sufficiently clear how environmental processes such as variations in hydrological state variables (water storage) or water
        fluxes may translate into the observation of the monitoring devices used here, i.e., gravimeters and tiltmeters. Given that the
265     hydrological community is hardly familiar with gravimeters, and even less with tiltmeters, large part of the interpretation of
        the monitoring data presented in this study remains unclear or inconclusive to the reader as the basic idea behind these
        instruments is not sufficiently laid out.
        Thus, I suggest to include in a revised version of the manuscript an introductory part that illustrates the measurement
        principle of gravimeter and tiltmeters and the influencing factors, and sets up general hypothesis how hydrological dynamics
270     might be seen by these instruments, probably also highlighting in which way the instruments react differently to the same
        process. On these grounds, in the results and discussion chapters of the manuscript, explaining and discussing the
        observations at the Conrad Observatory can then be more clearly presented in particular with respect to the following issues:
        - "Gravity and tilt residuals are associated to the same hydrological process but have different physical causes." (Abstract, ).
        What exactly are the physical causes that makes the difference between the instruments if the fundamental hydrological
275     process is the same?
        Reply:
        Thanks for this valuable suggestion. We completely re-organized the structure of the paper and explained in detail the
        different sensitivity of the used sensors to physical processes like gravitation and deformation and what they are able to
        detect. We also explained disturbing effects by strain-tilt coupling for the tiltmeters (cavity effect) and why this effect is
280     stronger at CO for the N-S tilt component. References to published results based on using gravimeters and tiltmeters in
        hydro-geological studies are now already provided in the introduction. We skipped technical details and shifted the details of
        the air pressure admittance study to an appendix for interested readers.

        - The "cavity effect" is mentioned in several instances throughout the manuscript as an influencing factor (abstract, line 177,
285     line 229, line 262) but it is not further explained. What is it about and how can it influence the observations? How can thus
        the statement "Because the tunnel axis is oriented in E-W direction, the N-S component corresponds to the tilt perpendicular
        to the tunnel axis and therefore is extremely sensitive to cavity effects." (line 177) be explained?
        - Additional explanations to the two points before may also shed more light on ": : :because tilt is affected by the topography
        and by geometry and size of the cavity where the tilt meters are installed" (line 33). This sentence is not intelligible by its
290     own for someone who is not familiar with tiltmeters.
        Reply to both comments:
        Re-organizing our paper offered opportunity to address the problem already in the introduction and to make it clear (please
        refer to our reply above).

295     - Line 227: "However, at long periods the air pressure signal in the tilt meter time series is due to geophysical/geodynamical
        reasons which are probably dominated by deformation due to air pressure loading." Also the gravimeter should be sensitive
        to loading effects that are associated with vertical displacements, right? Can this be jointly analyzed? More basically, even
        the term 'loading' might need to be explained for a hydrological reader.
        Reply:

300 Generally a gravimeter is sensitive to deformation effects as well: if deformation is associated with vertical displacement of the Earth's surface, then the gravimeter moves within the gravity field of the Earth and experiences different gravity. Generally, upward and downward movement decreases and increases gravity, respectively. Secondly, deformation always means mass redistribution and consequently a gravity change. However, at local spatial scale, vertical displacement generated by air pressure variation is very small and the associated gravity effect is negligible. At regional or global scale

305 such effects have to be considered. Another effect of the displacement is inertial acceleration if the displacement is time dependant (seismometer principle). However, this is important only at high frequencies (> 0.1 mHz). We explained this briefly.

- "Newtonian acceleration" Newtonian effect" (line 230, line 241) on the gravimeter needs to be explained with respect to

310 water storage (mass) variations. Also, what is the difference to the 'Newtonian tilt effect' (line 243) seen by tiltmeters?
Reply:
"Newtonian" is another wording meaning "gravitational". Any mass movement in the atmosphere and hydrosphere (air mass, water etc.) changes the gravitational potential and consequently the gravity field. Gravimeters are sensitive to the vertical component (or the norm) of the gravity vector while tiltmeters are sensitive to the horizontal component. We

315 explained this briefly in the introduction.

- Line 257: ": : : the observed total N-S tilt offsets as function of cumulative rain or the surface pressure load exerted by cumulative rain at the end of the respective rain event." Does a spatially uniform rain event cause a tilt signal? Probably not because also the surface pressure load is uniform? Thus, a tilt signal indicates spatially nonuniform rainfall?

320 Reply:
The gravitational effect of rain/snow water is too small to emerge out of the noise of the tiltmeters. However, like atmospheric pressure variations, also water at the surface exerts pressure onto the surface which results to deformation. Tiltmeters are sensitive to even little deformation and therefore experience a clear signal. This signal might be different when the load is uniform or non-uniform, but this cannot be addressed without further investigation and is out of the scope of

325 our paper.

- Line 262: "The short-term N-S tilt response is therefore interpretable as pure deformation effect (strain induced tilt) due to surface load, which is probably enhanced by the cavity effect." What does strain-induced tilt mean? How does this relate to the "cavity effect"?

330 Reply:
The cavity effect belongs to strain-induced tilt. You are right, our formulation is confusing, and we corrected this.

- Line 274: "Therefore, deformation due to surface loading rather than due to pore pressure changes explains the observed short-term tilt signal." This statement is not clear a another effect is introduced that has not been explained before: how and

335 why due pore pressure changes cause tilt signals? How do pore pressure changes relate to water storage changes that occur during a rainfall event?
Reply:
Water percolation into the subsurface or injection of water in boreholes is able to change the pore pressure which causes deformation. We addressed this in the introduction.

340
- Line 298: ": : : a clear systematic tendency of the source azimuth (340_ to 350_) is indicated." What does this mean? Needs some general introduction or explanation.
Reply:
Comparing the E-W and N-S tilt data, the amplitude ratio of the long-term residual anomalies turns out to be about−0.15 on

345 average; E-W tilt is always positive, N-S tilt is always negative. If the observed tilt is caused by gravitational attraction by a volume of stored water, then the source must be located on a line with azimuth of about 170 (the azimuth of 340 to 350° mentioned in the submitted MS was a misprint). We clarified this.

- Line 347: "It is not the physical source, but the hydrological process, which links the residual anomalies of gravity and tilt."
This statement is not clear. Is a hydrological process different from a physical source? What does this imply?
Reply: The hydrological process is water accumulation (producing short-term residual anomalies) or infiltration (producing long-term anomalies). The physical reason for the reaction of gravimeters is different from that of tiltmeters. Gravity residuals predominantly reflect the gravitational effect of water mass accumulation close to the surface (short-term) or water transport downwards (long-term). In contrast, the tiltmeters reflect the tilt due to deformation by the water load on topography (short-term) or due to deformation by water percolated into the subsurface (long-term). We think all this is more clearly described in the introduction and the discussion.

Other comments:
- Data section 2: I assume that there are no soil moisture or groundwater level data available at the observatory site or close to it? Is there a nearby river gauging station (or a smaller creek gauge) of which the discharge data could be used for comparing to the overall hydrological response of the study area?
Reply:
You are right. Unfortunately we do not have any hydrological instrumentation. There is river at the foot of Trafelberg mountain; however it would be unclear what amount of water in the creek stems from the Trafelberg massif or from other catchment areas around.

- The manuscript ends rather abruptly. I suggest adding a concluding paragraph on what has been learned from this combined setup of gravimeters and tiltmeters towards their potential for unraveling water storage dynamics and hydrological processes, what are the limitations, what are additional observations that may be needed to disentangle ambiguities in these observations, or similar aspects.
Reply:
We added a short paragraph on these aspects at the end.

**List of relevant changes:**

1. Complete re-organization of the text: We extended the introduction by explaining the physical background of tilts and how they originate on a deformable planet. We addressed the problem of strain-tilt coupling (cavity effect) in more detail and provided an overview on hydrology related papers on tilt observations already in the introduction.
2. Streamlining chapter 3 (Pre-processing, residual determination): We skipped the description of standard procedures and kept only those parts that are specific for the instrumentation at CO.
3. Streamlining chapter 4 (Tide models, air pressure admittance): we removed all technical details or shifted them to appendix A for interested readers.
4. Quantification of the relation between the gravity and tilt residuals and between the tilt components: we added this information to chapter 5.
5. Gravity residual modeling: We extended the discussion/conclusion chapter by presenting modeling results regarding the gravity response to hydrological processes observed at CO. Although these models have to be rather simplistic due to the lack of reliable constraints, they allow saturation estimates for the subsurface around the observatory.
6. General changes: We corrected unclear text formulations, in particular where requested by the reviewers, and tried to improve the language style.

[revised manuscript text omitted]
\phi[°]$ $\sigma(\varphi\phi)$ | $amp_{theor}$ [nrad] | $\gamma$ $\sigma(\gamma)$ | $\varphi\phi[°]$ $\sigma(\varphi\phi)$ | $\gamma$ $\sigma(\gamma)$ | $\varphi\phi[°]$ $\sigma(\varphi\phi)$ | |
| O1 | 0.6976 | 3.1275 | 1.0997 ±0.0358 | 7.235 ±1.867 | 23.4455 | 0.7135 ±0.0034 | −6.902 ±0.272 | 0.6746 ±0.0058 | −11.349 ±0.493 | 1.0577 |
| K1 | 0.7379 | 4.3962 | 1.2920 ±0.0259 | −1.318 ±1.152 | 32.9605 | 0.7773 ±0.0026 | −7.615 ±0.187 | 0.7278 ±0.0040 | −11.244 ±0.314 | 1.0680 |
| N2 | 0.6945 | 7.3013 | 0.6648 ±0.0067 | −2.691 ±0.579 | 9.8359 | 0.7673 ±0.0039 | −2.495 ±0.288 | 0.7130 ±0.0048 | −4.449 ±0.382 | 1.0762 |
| M2 | 0.6945 | 38.1330 | 0.6628 ±0.0013 | −3.193 ±0.115 | 51.3708 | 0.7401 ±0.0008 | −4.102 ±0.060 | 0.6849 ±0.0010 | −5.525 ±0.080 | 1.0806 |
| S2 | 0.6945 | 17.7398 | 0.6777 ±0.0031 | −2.880 ±0.264 | 23.8984 | 0.6896 ±0.0018 | −2.676 ±0.146 | 0.6203 ±0.0023 | −5.015 ±0.204 | 1.1117 |
| K2 | 0.6945 | 4.8195 | 0.6612 ±0.0129 | −2.531 ±1.116 | 6.4926 | 0.6880 ±0.0073 | −2.577 ±0.608 | 0.6385 ±0.0097 | −4.875 ±0.871 | 1.0775 |

**Table 1: Comparison of tidal parameters derived from tilt time series at CO. Theoretical body tide model: Dehant et al. (1999).**

| Air pressure admittance [nrad $hPa^{-1}$] | | |
|---|---|---|
| N-S | E-W | |
| LTS-Y | LTS-X | iWT-X |
| 4.247 ±0.034 | 0.097 ±0.019 | −0.475 ±0.034 |

**Table 2: Air pressure admittances in the diurnal and semidiurnal frequency band for the tilt sensors derived from tidal analysis.**

[Figure]

**Figure A1: Air pressure admittance function of the LTS/N-S tilt sensor derived from different observation periods covering a few days to less than 3 weeks each. Admittance (left), phase (middle), coherence (right). Circles and lines with intensive colours show the admittance (left), phase (middle) and coherence (right) respectively, averaged over the time series within 4 intervals (beginning - May 2017; May 2017 - June 2018/LTS repair; June 2018 - September 2018/LTS repair; September 2018 - end).**

[Figure]

**Figure A2: Air pressure to tilt (N-S) admittance function and their temporal evolution at selected frequencies derived by using data from different air pressure sensors (bottom: LTS air pressure sensor; middle: SG air pressure sensor). The top panel shows the SG to LTS air pressure admittance function.**

1355

---

## Referee Report (RR1)

[referee-annotated manuscript omitted]

---

## Author Response (AR2)

**Author's response**

We thank Michel van Camp for reviewing the revised manuscript. We first indicate our reaction to comments and suggestions by adding our replies and then present the manuscript allowing for tracing the changes.

Reviewer 1:

Suggestions for more precise formulations and language style improvements provided on following lines (numbering according to the annotated PDF of the reviewer):
26, 27, 41, 77, 84, 95, 132, 137, 240, 282, 377, 397, 417, 443
We considered all suggestions.

Line 73: We mentioned the paper by Tenze et al. in the discussion chapter and mention it now in the introduction additionally.

Line 179: "episodic temperature increase": We see the disturbances immediately after the light in the tunnel has switched on. This suggests heat transfer by radiation rather than by conduction to be the reason for temperature increase. Since the installation of insulation in August 2017 these disturbances do no longer happen proving the effectiveness of the insulation. We did not add an explanation to the text as we think this is too specific.

Line 186: "This happens during large earthquakes…": It causes offsets just during an EQ which have to be corrected for. No data are lost in contrast to SG records during very large (M >=7) events. After 2-10 minutes instrument is capable to record again tilting. It is necessary interpolate the gap as usual in earth tides recordings. Earthquakes larger than M=7.3 produce steps in the data set. The earthquakes lower or equal than M=7.3 cause offsets if the hypocenter is shallow. EQ with $6.8 \leq M < 7.0$ produce offsets if their hypocenters are situated near CO (longitude difference <50°). EQ with $6.5 \leq M < 6.8$ usually do not cause steps in the data set. EQ with M < 6.5 produce steps, if the earthquake epicentre is near to CO (< 15°). We did not add an explanation to the text as we think all this is too specific.

Line 247: We added a remark on the rain admittance concept as suggested by the reviewer.

Line 295/296: Yes, we think it is the effect of pressure exerted onto the ground, at least regarding the immediate response to rain Water/snow accumulation on ground which we refer to.

Line 322: we split up the chapter "Discussion and Conclusion" in two chapters "Discussion" and "Conclusion".

Line 463: "After first repair": The reason is unclear. Contrary to the 2nd repair, the tiltmeter box has not been opened because only electronic damage had to be repaired. Therefore, we suppose that the sensor is sensitive to transport vibrations. We did not add an explanation to the text as this is speculative and would require more detailed investigation, out of the scope of our paper.

Line 464: "coherence": We added an explaining statement.

[revised manuscript text omitted]